# Breaking Manifold Continuity: Vector Quantized Modeling for Real-Centric Deepfake Detection

Changshuo Wang [* 1 2]   Jiangming Wang [* 2]   Ke-Yue Zhang [2]   Taiping Yao [2]   Shouhong Ding [2]   Ran Yi [1]
Lizhuang Ma [1]

## Abstract

The increasingly realistic and diverse generative data has led some deepfake detection methods to shift towards learning robust real content, *e.g.*, via reconstruction-based tasks. However, most existing approaches rely primarily on prevalent continuous modeling (*e.g.*, GMMs, VAEs, Diffusion Models) to construct a continuous latent manifold of real data, with the aim of improving the generalization capability, while overlooking a critical issue, *i.e.*, such continuity may facilitate the interpolation of forgery artifacts, consequently causing ambiguity in detection. To alleviate this problem, we integrate discrete modeling into the feature space of the CLIP vision encoder, striking a balance between continuous manifold modeling and discrete representation. By incorporating a learnable vector quantized codebook, the real latent manifold is discretized, imposing a more stringent information bottleneck that reduces the likelihood of embedding generative artifacts. In order to further enhance the generalization of discrete modeling, we propose an adaptive tangent space projection mechanism that yields a continuous relaxation of the discrete real distribution within a controllable range. With these components, our method constructs a real distribution that is both tightly constrained and broadly generalizable, enhancing robustness to unseen forgeries. Extensive experiments on diverse datasets demonstrate the effectiveness of our method.

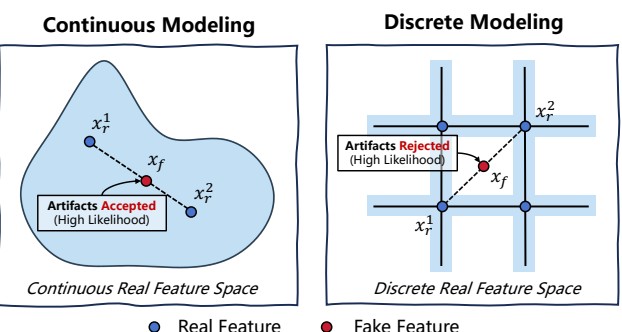

*Figure 1.* **Continuous vs. discrete modeling. Left:** Continuous modeling induces a smooth real-manifold feature space, where the fake ($x_f = \mathrm{Interp}(x_r^1, x_r^2)$) retains high likelihood of being interpolated in it, resulting in accepted artifacts. **Right:** Discrete modeling restricts the feature space to a grid-like structure, so the fake has a high likelihood of landing in the vacant cells of the "grid", leading to effective rejection.

## 1. Introduction

Recent progresses in generative models have made facial manipulations strikingly close to real images, underscoring the urgent need for reliable detection approaches. A majority of existing detection methods (Yan et al., 2023a; 2024; Ma et al., 2025; Yan et al., 2025b; Shiohara & Yamasaki, 2022) focus on learning forgery traces, which often leads to overfitting and poor generalization. Therefore, some advanced methods (Larue et al., 2023; Zou et al., 2025; Yan et al., 2025a) draw on anomaly detection paradigms to model real data. Instead of chasing the infinite variety of forgery patterns, these methods concentrate exclusively on learning generalizable real distribution, aiming to reject fake samples that deviates from the learned real manifold.

Nevertheless, these prevailing real-centric approaches solely rely on continuous reconstruction learning (*e.g.*, Variational Autoencoders (VAEs) (Kingma & Welling, 2013), Gaussian Mixture Models (GMMs) (McLachlan et al., 2019), Diffusion Models (Ho et al., 2020)), or continuous subspace learning (*e.g.*, LoRA (Hu et al., 2022), SVD mapping), ignoring a critical limitation of continuous modeling shown in Figure 1, *i.e.*, **the inherent continuity of the real latent space may enable smooth interpolation of forgery artifacts**,

---

[*]Equal contribution [1]Shanghai Jiao Tong University, Shanghai, China [2]Tencent Youtu Lab, Shanghai, China. Correspondence to: Ke-Yue Zhang <zkyezhang@tencent.com>, Lizhuang Ma <ma-lz@cs.sjtu.edu.cn>, Ran Yi <ranyi@sjtu.edu.cn>.

*Proceedings of the 43rd International Conference on Machine Learning*, Seoul, South Korea. PMLR 306, 2026. Copyright 2026 by the author(s).

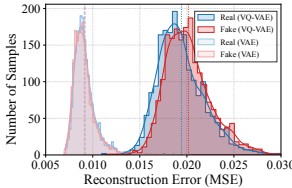 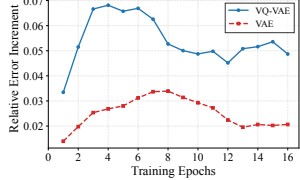

*(a)* MSE Distribution      *(b)* Relative Error Increment

*Figure 2.* **Discretizing the real latent manifold can amplify forgery artifacts in the image reconstruction space**, exemplified by VAE and VQ-VAE. We train models on real data of FF++ (Rossler et al., 2019) and evaluate them on its real and fake subsets. **(a)** The sample distribution of MSE Reconstruction Error ($\epsilon_{mse}^{real}$, $\epsilon_{mse}^{fake}$). **(b)** Evolution of the average Relative Error Increment ($\epsilon_{mse}^{fake}/\epsilon_{mse}^{real} - 1$) during training.

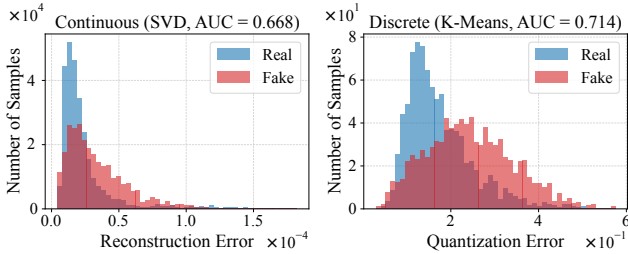

*Figure 3.* **Discrete real latent transformation is more discriminative than continuous one for real-centric forgery detection**, exemplified by SVD and K-Means. We train Effort (Yan et al., 2025a) on FF++ (Rossler et al., 2019) and evaluate on CDF-v2 (Yuezun Li & Lyu, 2020) without classifier. **Left:** Continuous real latent mapping ($V^T V$) via SVD (top 95% components) with feature-level MSE Reconstruction Error and AUC. **Right:** Discrete real latent mapping via K-Means (K=2048) with feature-level Quantization Error (L2 to nearest real prototype) and AUC.

consequently **leading to ambiguity in detection**. To gain deeper insight into this limitation, we perform a sequence of preliminary experiments, progressing from the real/fake image reconstruction space to the real/fake latent discrimination space. As illustrated in Figure 2, we first compare the real and fake reconstruction performance of VAE and VQ-VAE under a controlled setting where only real data are used for training. We not only plot the sample distribution of MSE Reconstruction Error ($\epsilon_{mse}^{real}$, $\epsilon_{mse}^{fake}$) but also define the average Relative Error Increment ($\epsilon_{mse}^{fake}/\epsilon_{mse}^{real} - 1$) to quantify their discrepancy during training. Notably, **VAE exhibits weaker real/fake separability than VQ-VAE with respect to the MSE Reconstruction Error**, which means that enforcing discreteness on the real latent manifold amplifies forgery artifacts in the image reconstruction space. In Figure 3, we further conduct experiments directly aligned with the feature-level forgery detection by modeling real data. Specifically, we apply SVD (top 95% components) on the real latent space to create a continuous latent transformation ($V^T V$) with right-singular matrix and apply K-Means (K=2048) on the real latent space to create a discrete latent transformation (cluster centroids as prototypes). We then compute feature-level MSE reconstruction error for SVD and quantization error (L2 distance to the nearest prototype) for K-Means, and evaluate AUC based on these errors. **The results show that SVD yields a less separable real–fake error distribution than K-Means**, suggesting that discrete latent modeling may have an inherent advantage over continuous latent modeling for this discriminative task.

Inspired by the above results and the real-centric learning paradigm, we propose a two-stage training framework for real-centric deepfake detection task via **integrating discretization into continuity**. In the first stage, we establish a robust real distribution by incorporating a learnable real codebook with carefully designed vector-quantization losses, which discretize the CLIP vision encoder's latent feature space and thereby reduce the likelihood of interpolating forgery artifacts. This effect arises because the upper bound

of the mutual information $I(X; Z_{disc})$ between the input $X$ and the discrete latent $Z_{disc}$ is empirically tighter than the mutual information $I(X; Z_{cont})$ between $X$ and the continuous latent $Z_{cont}$ (proved in Section 3). As a result, when reconstructing the same content, the discrete latent representation can force encoder to discard more spurious information that could promote interpolated artifacts, and instead retain the most informative features for distinguishing real data from forgeries. To further overcome generalization problem of discrete real modeling, we develop an adaptive tangent space projection mechanism to reintroduce local controllable continuity. By employing the masked modeling with tangent space constraint, we strike a balance between continuity and discretization, yielding the discriminative reconstruction residuals for real/fake classification.

Our main contributions are summarized as follows:

- **Limitation of continuous real-centric modeling:** We identify the overlooked limitation in real-centric deepfake detection, *i.e.*, continuous real modeling can promote the interpolation of forgery artifacts, leading to ambiguous detection.

- **New perspective of discrete modeling as an information bottleneck:** We investigate this limitation both theoretically and experimentally, showing that incorporating discrete modeling serves as a strict latent information bottleneck that effectively alleviates it.

- **Novel discrete-enhanced real-centric framework:** We incorporate the discretization into continuity and propose a novel real-centric framework that combines real vector-quantized codebook learning with an adaptive tangent space projection mechanism, producing discriminative reconstruction residuals that enhance forgery detection performance.

## 2. Related Work

### 2.1. Generalizable Deepfake Image Detection

Generalization remains a central challenge in deepfake image detection. Existing methods can be broadly categorized into CNN-based and ViT-based approaches. Early works predominantly adopt CNN architectures (e.g., Xception (Chollet, 2017)) due to their strong inductive bias for capturing local low-level artifacts. To improve generalization, these methods typically rely on extensive data augmentation, including frequency-domain transformations (Li et al., 2021; Luo et al., 2021; Liu et al., 2021), blending-based synthesis (Li et al., 2020b; Zhao et al., 2021; Shiohara & Yamasaki, 2022; Chen et al., 2022), and identity decomposition strategies (Yan et al., 2023a; Huang et al., 2023; Dong et al., 2023). Despite their effectiveness on seen attacks, such supervised binary classifiers often fail to generalize to unseen generation methods.

Recently, several approaches (Fu et al., 2025; Cui et al., 2025; Ma et al., 2025; Yan et al., 2025a) leverage pre-trained vision foundation models (e.g., CLIP (Radford et al., 2021), DINO (Siméoni et al., 2025)) to enhance generalization. These ViT-based models have a great pre-trained real knowledge learned from large-scale real data. However, most existing methods rely on continuous manifold modeling, which, while effective for capturing semantic variations, may inadvertently allow high-frequency forgery artifacts to remain undetected.

### 2.2. Real-Centric AI-Generated Image Detection

To mitigate the overfitting risks of binary classifiers, recent research has shifted towards One-Class Learning, modeling the authentic distribution exclusively to identify anomalies. Generative approaches like OC-FakeDect (Khalid & Woo, 2020) utilize VAEs to detect forgeries via reconstruction errors, while feature-level methods (Zou et al., 2025) employ Gaussian Mixture Models (GMM) to approximate the probability density of real features. Similarly, Ojha et al. (Ojha et al., 2023a) leverage the feature space of CLIP to capture universal artifacts across different generative models. However, both VAEs and GMMs fundamentally rely on continuous manifold modeling. Consequently, recent works have shifted towards discrete representations. Prototype-based methods like SeeABLE (Larue et al., 2023) map samples to fixed anchors to generate pseudo-forgeries. Similarly, D3QE (Zhang et al., 2025) explores the discrete nature of autoregressive models, utilizing quantization errors and codebook frequency biases for detection. While these approaches introduce discrete concepts, they often rely on fixed prototypes or specific generative statistics, limiting the model's capacity to adaptively capture the complex geometric variations of authentic faces.

## 3. IB Perspective: Continuous vs. Discrete

We provide an intuitive explanation, from the perspective of Information Bottleneck (IB) theory, of why a discrete latent space inherently enforces a much tighter information bottleneck than a continuous one under reconstruction view. Let $X$ be the input random variable, and let $Z$ be its latent representation obtained via an encoder $q_\phi(z|x)$, where $x$ and $z$ denote a specific sample draw from $X$ and $Z$, respectively.

**For a continuous latent** $Z_{cont}$, the mutual information is defined as:

$$I_\phi(X; Z_{cont}) = h_\phi(Z_{cont}) - h_\phi(Z_{cont}|X) \to \infty, \quad (1)$$

where $h_\phi(\cdot)$ is differential entropy with the definition as $h_\phi(Z) = -\int p_\phi(z)\log(p_\phi(z))dz$. Notably, $h(Z)$ can grow without bound when a continuous encoder allocates more capacity, so with weak regularization (*e.g.*, in a standard VAE), the effective upper bound on $I(X; Z_{cont})$ becomes loose and very large, allowing the latent representation to retain excessive detail for reconstructing $X$.

**For a discrete latent** $Z_{disc}$, its latent space is quantized into a finite set (*e.g.*, codebook) $\mathcal{C} = \{e_k\}_{k=1}^K$. The mutual information is defined as:

$$I(X; Z_{disc}) = H(Z_{disc}) - H(Z_{disc}|X), \quad (2)$$

where $H(\cdot)$ is Shannon entropy with the definition as $H(Z) = -\sum_z p(z)\log(p(z))$. Since $Z_{disc}$ has at most $K$ states, we can deduce:

$$I(X; Z_{disc}) \leq H(Z_{disc}) \leq \log_2 K. \quad (3)$$

Thus, the discrete latent has a controllable upper bound that mainly depends on the number of latent states $K$. Compared to the continuous latent, this tighter information bottleneck encourages discarding redundant details and retaining the most informative factors of $X$.

## 4. Methodology

The overall training framework, comprising Discrete Codebook Learning and the Adaptive Tangent Space Projection Mechanism, is illustrated in Figure 4, while the inference process is shown in Figure 5. These two training stages mutually reinforce each other to construct a more generalizable real distribution, thereby enabling reconstruction residuals to serve as reliable cues for forgery detection.

### 4.1. Problem Setting

Let $\mathcal{D} = \{x_n, y_n\}_{n=1}^N$ be a certain dataset, where $x_n \in \mathbb{R}^{H \times W \times 3}$ denotes the input data, and $y_n \in \{0, 1\}$ denotes the corresponding label (0 for real, 1 for fake). The goal of this problem is to learn a model that can reliably distinguish real images from fake ones. To simplify the notation, we omit the subscript $n$ when it is clear from the context.

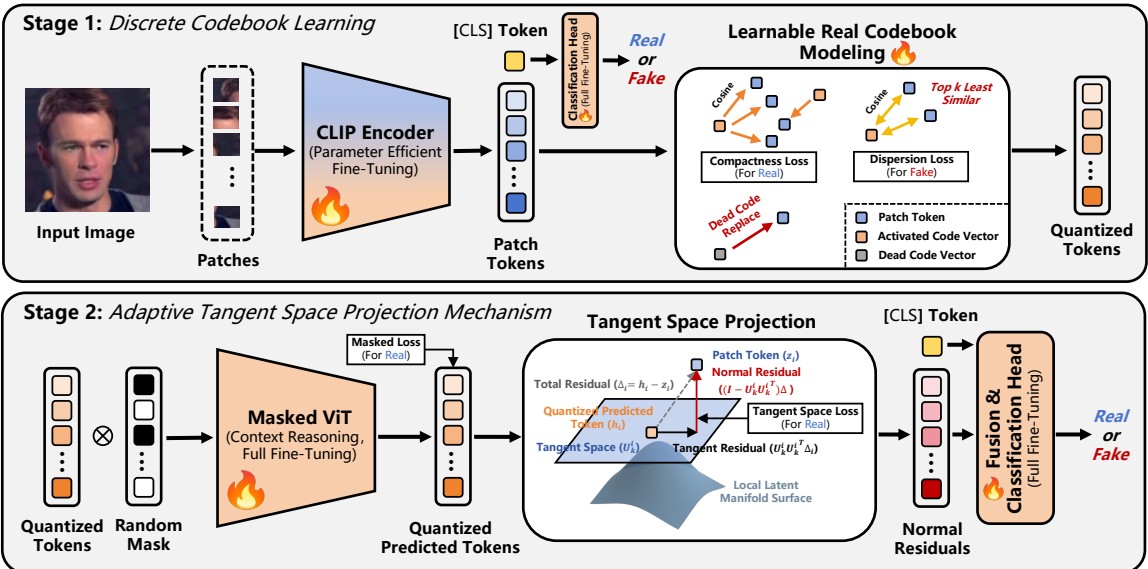

*Figure 4.* **Overview of our training framework.** Face images used are from CDF-v2 ([Yuezun Li & Lyu](), 2020). The training proceeds in two progressive stages. **Stage 1**: A CLIP encoder with parameter efficient fine-tuning is employed to project real patch tokens onto a learnable codebook, discretizing the latent manifold to establish real distribution. **Stage 2**: A masked ViT first restores contextual dependencies among quantized tokens. Next, tangent space projection models the local linear subspace around each code vector, and a hypernetwork then fuses global semantic information with local geometric residuals to perform forgery detection.

### 4.2. Discrete Codebook Learning

Our primary objective in this first stage is to construct a discrete support set (the codebook) $\mathcal{C}$ that approximates the intrinsic latent manifold of the real data.

**Feature Space Adaptation.** Instead of training from scratch, we leverage the semantic priors of the pre-trained CLIP model. To adapt the general visual representations to the specific domain of face forensics without catastrophic forgetting, we employ Effort ([Yan et al., 2025a]()) on the CLIP vision encoder $E_\theta(\cdot)$. Given an input $x$, we extract the sequence of patch-level tokens $Z \in \mathbb{R}^{L \times D}$:

$$Z = E_\theta(x) = \{z_1, z_2, ..., z_L\}, \quad (4)$$

where $L$ is the number of patches and $D$ is the embedding dimension.

**Discretized Manifold Anchoring.** We introduce a learnable discrete codebook $\mathcal{C} = \{e_k\}_{k=1}^K \in \mathbb{R}^{K \times D}$, where $K$ is the codebook size. This codebook serves as a set of anchors discretizing the continuous feature space into a Voronoi tessellation. For each patch token $z_i$, we obtain a quantized token set $Z_q = \{z_q^1, ..., z_q^L\}$ by mapping it to its nearest neighbor in the codebook:

$$z_q^i = e_k, \quad \text{where } k = \underset{k \in 1, ..., K}{\arg\min} ||z_i - e_k||_2. \quad (5)$$

The codebook and encoder are trained under a dual-objective mechanism. **For real samples**, we employ the

standard VQ objective to encourage the patch tokens to cluster tightly around their related code vectors. This includes the quantization loss (updating the codebook) and the commitment loss (updating the encoder):

$$\mathcal{L}_{\text{COM}} = ||\text{sg}[Z] - Z_q||_2^2 + \beta ||Z - \text{sg}[Z_q]||_2^2, \quad (6)$$

where $\text{sg}[\cdot]$ denotes the stop-gradient operator, and $\beta$ is a hyperparameter balancing the encoder's commitment. **For fake samples**, we introduce a margin-based dispersion loss $\mathcal{L}_{\text{DIS}}$. Since the codebook is designed to represent only real patterns, embeddings of fake patches containing artifacts should be pushed away from their nearest codebook entries to prevent negative matching:

$$\mathcal{L}_{\text{DIS}} = \frac{1}{K} \sum_{j \in \mathcal{T}} \max(0, \xi - ||z_j^{fake} - z_q^j||_2), \quad (7)$$

where $\xi$ is a predefined distance margin and $\mathcal{T}$ denotes an indexed set. Since deepfake artifacts are often spatially localized, averaging the loss over all $L$ patches can dilute forgery signals. Therefore, we select a subset $\mathcal{T}$ of the top-$K$ patches with the largest quantization errors and penalize the model only when these most anomalous fake features lie too close to the real manifold.

The total batch-level objective for Stage 1 is formulated as:

$$\mathcal{L}_{\text{Stg1}} = \frac{1}{N_b} \sum_{n=1}^{N_b} \mathcal{L}_{\text{CE}}^{(n)} + \frac{1}{N_b^r} \sum_{n=1}^{N_b^r} \mathcal{L}_{\text{COM}}^{(n)} + \frac{1}{N_b^f} \sum_{n=1}^{N_b^f} \mathcal{L}_{\text{DIS}}^{(n)}, \quad (8)$$

where $N_b, N_b^r, N_b^f$ denotes the number of total, real, fake images in a batch, respectively, and $\mathcal{L}_{\text{CE}}$ denotes the

real/fake binary classification loss. By optimizing this objective, the codebook $\mathcal{C}$ is refined into a compact set of discrete manifold anchors, such that real samples exhibit minimal quantization error, whereas fake samples retain substantial quantization residuals.

### 4.3. Adaptive Tangent Space Projection Mechanism

While the codebook effectively rejects interpolated artifacts, its discrete and independent nature may sacrifice certain generalization capabilities, *e.g.*, handling subtle illumination changes or pose shifts. To solve it, we introduce the masked modeling followed by a tangent space projection in the second training stage. Notably, we freeze the parameters of the encoder and the codebook learned in the first stage.

**Contextual Restoration via Masked Modeling.** The codebook quantization processes each patch independently, which ignores the direct relationships among patches. To deal with it, we employ a masked ViT to restore spatial dependencies. Given the sequence of quantized tokens $Z_q$, we randomly mask 75% of them with a special mask token, and feed the resulting sequence into a ViT model to obtain the context-aware quantized predicted tokens $H = \{h_1, ..., h_L\}$. The learning objective is to reconstruct the original latent representations of the masked ones:

$$\mathcal{L}_{\text{MIM}} = \frac{1}{|\mathcal{M}|} \sum_{i \in \mathcal{M}} ||h_i - z_i||_2^2, \tag{9}$$

where $\mathcal{M}$ denotes the set of masked indices. Notably, this loss is only applied to real samples. After that, the quantized predicted tokens capture context-enhanced real information and further improve the robustness of real latent distribution.

**Tangent Space Projection.** Based on the manifold hypothesis, we assume that for each code vector $e_k \in \mathcal{C}$, the associated real latent features are distributed on a local low-dimensional manifold around it, rather than collapsing to a single point (empirically validated in Figure 8). Since $h_i$ is derived from its code vector $e_k$, it follows the same manifold hypothesis and lies in the local neighborhood around $e_k$. We approximate this local geometry by the tangent space. Formally, we learn a specific projection matrix $U_k \in \mathbb{R}^{D \times d}$ (where $d \ll D$) that spans the principal subspace (*i.e.*, the tangent plane) of the manifold for each $e_k$. The columns of $U_k$ form an orthogonal basis, *i.e.*, $U_k^T U_k = I_d$. Therefore, for each patch token $z_i$, we can decompose its residual vector $\Delta_i = h_i - z_i$ into two orthogonal components ($U_k^i$ denotes the projection matrix of $e_k$ that is activated by $z_i$):

- **Tangent Residual** ($\Delta_i^{\parallel} = U_k^i U_k^{i^T} \Delta_i$): The projection onto the tangent space, representing valid continuous variations.

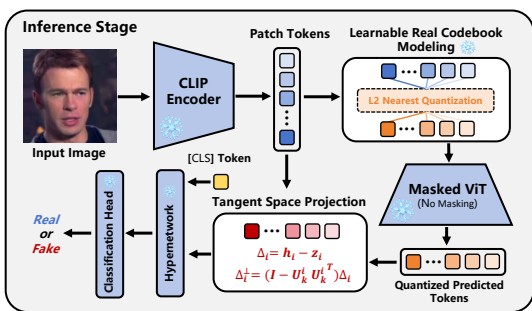

*Figure 5.* **Overview of our inference framework.** Face images used are from CDF-v2 (Yuezun Li & Lyu, 2020).

- **Normal Residual** ($\Delta_i^{\perp} = (I - U_k^i U_k^{i^T})\Delta_i$): The projection onto the orthogonal complement, representing deviations that leave the latent manifold.

To learn robust real tangent space, we minimize the normal residuals. This objective encourages the restored feature $h_i$ to lie close to the subspace spanned by $U_k^i$:

$$\mathcal{L}_{\text{TSP}} = \frac{1}{L} \sum_{i=1}^{L} \left( ||(I - U_k^i U_k^{i^T})\Delta_i||_2^2 + ||U_k^{i^T} U_k^i - I||_F^2 \right), \tag{10}$$

where $|| \cdot ||_F$ denotes the Frobenius norm. The second term in loss imposes an orthogonality constraint to ensure that $U_k$ forms a valid orthonormal basis of the corresponding subspace. Notably, the loss is enforced exclusively on $\Delta_i$ derived from real samples.

**Residual Fusion Learning.** Finally, we concatenate all normal residuals to form a residual matrix $\mathcal{R}_n = \text{Concat}([\Delta_1^{\perp}, ..., \Delta_L^{\perp}]) \in \mathbb{R}^{L \times D}$ and introduce a hypernetwork $\Psi$ allowing the [CLS] token $z_{cls}$ to modulate the importance of residuals. In particular, $\Psi$ maps $z_{cls}$ to a sample-specific weight vector $W \in \mathbb{R}^{1 \times L}$, which is applied to $\mathcal{R}$ together with a shared learnable bias $b^{\star}$ to obtain a weighted residual. The weighted residual is then fed into a classification head $F(\cdot)$ to compute discrimination loss:

$$\mathcal{L}_{\text{CLS}} = \mathcal{L}_{\text{CE}}(F(W \odot \mathcal{R} + b^{\star}), y). \tag{11}$$

The total batch-level objective for Stage 2 is formulated as:

$$\mathcal{L}_{\text{Stg2}} = \frac{1}{N_b^r} \sum_{n=1}^{N_b^r} \left( \mathcal{L}_{\text{MIM}}^{(n)} + \lambda_{\text{TSP}} \mathcal{L}_{\text{TSP}}^{(n)} \right) + \frac{1}{N_b} \sum_{n=1}^{N_b} \mathcal{L}_{\text{CLS}}^{(n)}, \tag{12}$$

where $\lambda_{\text{TSP}}$ is a hyperparameter to control the importance of $\mathcal{L}_{\text{TSP}}$. By optimizing this objective, we can effectively leverage reconstruction anomalous residuals to learn a more generalizable model.

### 4.4. Inference Details

For inference stage, the masking operation is disabled so that the model processes the entire image. The frozen CLIP encoder first extracts patch tokens $Z$ and the [CLS] token $z_{cls}$

*Table 1.* **Cross-dataset evaluation results.** All models are trained on FF++ (c23) (Rossler et al., 2019), and we present video-level AUC scores on various test sets. Results marked with an asterisk (*) are our reproductions using the official code under consistent settings. The dagger symbol (†) indicates results taken from the original publications, and the remaining results are from (Yan et al., 2025a; 2023b).

| Methods | Venue | CDF-v2 | FSh | DFD | DFDC | DFDCP | WDF | Avg. |
|---|---|---|---|---|---|---|---|---|
| F3Net (Qian et al., 2020) | AAAI'20 | 0.789 | 0.638 | 0.844 | 0.718 | 0.749 | 0.728 | 0.744 |
| SRM (Luo et al., 2021) | CVPR'21 | 0.840 | 0.729 | 0.885 | 0.695 | 0.728 | 0.702 | 0.763 |
| RECCE (Cao et al., 2022) | CVPR'22 | 0.823 | 0.801 | 0.891 | 0.696 | 0.734 | 0.756 | 0.784 |
| IID (Huang et al., 2023) | CVPR'23 | 0.838 | 0.850 | 0.939 | 0.700 | 0.689 | 0.666 | 0.780 |
| UCF (Yan et al., 2023a) | ICCV'23 | 0.837 | 0.879 | 0.867 | 0.742 | 0.770 | 0.774 | 0.812 |
| SeeAble (Larue et al., 2023) | ICCV'23 | 0.873 | - | - | 0.759 | 0.863 | - | - |
| LSDA (Yan et al., 2024) | CVPR'24 | 0.875 | 0.856 | 0.881 | 0.701 | 0.812 | 0.797 | 0.820 |
| SBI (Shiohara & Yamasaki, 2022) | CVPR'22 | 0.886 | 0.839 | 0.827 | 0.717 | 0.848 | 0.703 | 0.803 |
| ProDet (Cheng et al., 2024) | NeurIPS'24 | 0.926 | 0.818 | 0.901 | 0.707 | 0.828 | 0.781 | 0.827 |
| SLADD (Chen et al., 2022) | CVPR'22 | 0.837 | 0.793 | 0.904 | 0.772 | 0.756 | 0.690 | 0.792 |
| CDFA (Lin et al., 2024) | ECCV'24 | 0.938 | 0.836 | 0.954 | 0.830 | 0.881 | 0.796 | 0.873 |
| Effort† (Yan et al., 2025a) | ICML'25 | **0.956** | 0.868 | 0.965 | 0.843 | 0.909 | 0.848 | 0.898 |
| Forensics Adapter* (Cui et al., 2025) | CVPR'25 | **0.956** | 0.803 | 0.960 | 0.869 | 0.851 | 0.890 | 0.888 |
| DeepShield† (Yinqi et al., 2025) | ICCV'25 | 0.922 | - | 0.961 | 0.828 | **0.932** | - | - |
| VB† (Yan et al., 2025b) | CVPR'25 | 0.947 | - | 0.965 | 0.843 | 0.909 | 0.848 | - |
| FakeSTormer† (Dat et al., 2025) | ICCV'25 | 0.924 | - | **0.985** | 0.746 | 0.900 | 0.742 | - |
| Ours | - | 0.945 | **0.918** | 0.966 | **0.879** | **0.932** | **0.918** | **0.929** |

from input image $x$. Then, $Z$ is quantized to the nearest code vector and refined by the unmasked ViT model, yielding the quantized predicted tokens $Z_q$. After that, we calculate the normal residual matrix $\mathcal{R}$ by projecting features onto the orthogonal complement of the code vector's tangent space. Simultaneously, $z_{cls}$ drives the hypernetwork to generate sample-specific weighted matrix $W$. Finally, the classification score is computed as $p(x) = \text{Softmax}(F(\mathcal{R}; W, b^\star))$.

# 5. Experiment

## 5.1. Settings

**Evaluation Datasets and Metrics.** We train our models on FaceForensics++ (FF++) and evaluate their generalization performance on multiple widely used deepfake detection benchmarks, including Celeb-DF-v2 (CDF-v2) (Yuezun Li & Lyu, 2020), FaceShifter (FSh) (Li et al., 2020a), DeepfakeDetection (DFD) (DFD., 2020), DFDC (detection challenge., 2020), DFDCP (Dolhansky et al., 2019), and WDF (Zi et al., 2020). These datasets cover diverse manipulation methods, visual artifacts, and real-world scenarios, enabling a comprehensive evaluation of model robustness. Following prior works (Yan et al., 2025a; Cui et al., 2025; Yan et al., 2023a; 2024; Li et al., 2025; Yinqi et al., 2025; Dat et al., 2025; Zhaoyang et al., 2025), we report the widely-used video-level Area Under the Curve (AUC) as the primary evaluation metric. Specifically, we compute the average model's output probabilities of each video to obtain the video-level AUC.

**Implementation Details.** Our implementation is based on the training pipeline of DeepfakeBench (Yan et al., 2023b).

We adopt CLIP ViT-L/14 (Radford et al., 2021) as the vision foundation model. Following common practice (Yan et al., 2025a), we uniformly sample 8 frames per video during training and 32 frames per video for inference. Standard data augmentations, including Gaussian Blur and Image Compression, are applied during training (Yan et al., 2025a; Cheng et al., 2024; Shiohara & Yamasaki, 2022). The model is optimized with Adam (Kingma & Ba, 2014) (learning rate of $2 \times 10^{-4}$, batch size of 32). For the loss function, we set $\xi = 0.2$, $\lambda_{\text{TSP}} = 0.1$. Unless otherwise noted, all experiments share the same training and evaluation settings.

## 5.2. Generalization Evaluation

To comprehensively assess the generalization of our proposed method, we conduct experiments against a broad range of state-of-the-art detectors, including F3Net (Qian et al., 2020), SRM (Luo et al., 2021), RECCE (Cao et al., 2022), IID (Huang et al., 2023), UCF (Yan et al., 2023a), SeeAble (Larue et al., 2023), LSDA (Yan et al., 2024), SBI (Shiohara & Yamasaki, 2022), ProDet (Cheng et al., 2024), SLADD (Chen et al., 2022), CDFA (Lin et al., 2024), Effort (Yan et al., 2025a), Forensics Adapter (Cui et al., 2025), DeepShield (Yinqi et al., 2025), VB (Yan et al., 2025b), and FakeSTormer (Dat et al., 2025). Table 1 shows the **Cross-Dataset Evaluation** with video-level AUC scores on ten diverse test datasets covering different sources and visual styles. This test challenges the models to handle significant domain shifts and unknown manipulation techniques. Our method achieves the best performance on most datasets and attains the highest average video-level AUC by a clear margin, proving strong resistance to domain shifts. Table 2 presents the **Cross-Method Evaluation**, where the

*Table 2.* **Cross-method evaluation results.** All models are trained on FF++ (c23) (Rossler et al., 2019), and we present video-level AUC scores on various test sets. Results marked with an asterisk (*) were reproduced using the official code under the same experimental settings as other models, while all other results are sourced from (Yan et al., 2025a; 2023b).

| Methods | Venue | UniFace | BleFace | FaceDan | FSGAN | InSwap | SimSwap | Avg. |
|---|---|---|---|---|---|---|---|---|
| F3Net (Qian et al., 2020) | AAAI'20 | 0.809 | 0.808 | 0.717 | 0.845 | 0.757 | 0.674 | 0.768 |
| SRM (Luo et al., 2021) | CVPR'21 | 0.749 | 0.704 | 0.659 | 0.772 | 0.793 | 0.694 | 0.729 |
| RECCE (Cao et al., 2022) | CVPR'22 | 0.898 | 0.832 | 0.848 | 0.949 | 0.848 | 0.768 | 0.857 |
| SPSL (Liu et al., 2021) | CVPR'21 | 0.747 | 0.748 | 0.666 | 0.812 | 0.643 | 0.665 | 0.714 |
| IID (Huang et al., 2023) | CVPR'23 | 0.839 | 0.789 | 0.844 | 0.927 | 0.789 | 0.644 | 0.805 |
| UCF (Yan et al., 2023a) | ICCV'23 | 0.831 | 0.827 | 0.862 | 0.937 | 0.809 | 0.647 | 0.819 |
| LSDA (Yan et al., 2024) | CVPR'24 | 0.872 | 0.875 | 0.721 | 0.939 | 0.855 | 0.793 | 0.843 |
| SBI (Shiohara & Yamasaki, 2022) | CVPR'22 | 0.724 | 0.891 | 0.594 | 0.803 | 0.712 | 0.701 | 0.738 |
| ProDet (Cheng et al., 2024) | NeurIPS'24 | 0.908 | **0.929** | 0.747 | 0.928 | 0.837 | 0.844 | 0.866 |
| SLADD (Chen et al., 2022) | CVPR'22 | 0.878 | 0.882 | 0.825 | 0.943 | 0.879 | 0.794 | 0.867 |
| CDFA (Lin et al., 2024) | ECCV'24 | 0.762 | 0.756 | 0.803 | 0.942 | 0.772 | 0.757 | 0.799 |
| Forensics Adapter* (Cui et al., 2025) | CVPR'25 | 0.942 | 0.872 | 0.941 | 0.968 | 0.946 | 0.914 | 0.931 |
| Effort (Yan et al., 2025a) | ICML'25 | 0.962 | 0.873 | 0.926 | 0.957 | 0.936 | 0.926 | 0.930 |
| Ours | - | **0.979** | 0.923 | **0.953** | **0.970** | **0.960** | **0.937** | **0.954** |

*Table 3.* **Ablation study of key components on DFDC and DFDCP datasets.** We report the video-level AUC. The results show a consistent performance improvement with the addition of each module, confirming their complementary effectiveness.

| Ours | | | | DFDC | DFDCP | Avg. |
|---|---|---|---|---|---|---|
| Codebook | MViT | TSP | Fusion | | | |
| ✗ | ✗ | ✗ | ✗ | 0.843 | 0.909 | 0.876 |
| ✓ | ✗ | ✗ | ✗ | 0.861 | 0.918 | 0.890 |
| ✓ | ✓ | ✗ | ✗ | 0.867 | 0.920 | 0.894 |
| ✓ | ✓ | ✓ | ✗ | 0.873 | 0.927 | 0.900 |
| ✓ | ✓ | ✓ | ✓ | 0.879 | 0.932 | 0.906 |

*Table 4.* **Performance impact of codebook size $K$.** We report the video-level AUC on three datasets. The results indicate that $K = 1024$ achieves an optimal balance between the expressiveness of real features and the fidelity of transmitted information.

| Codebook Size ($K$) | Fsh | DFDC | DFDCP | Avg. |
|---|---|---|---|---|
| 128 | 0.852 | 0.836 | 0.892 | 0.860 |
| 256 | 0.898 | 0.862 | 0.915 | 0.892 |
| 512 | 0.910 | 0.871 | 0.932 | 0.904 |
| 1024 | 0.918 | 0.879 | 0.932 | 0.910 |
| 2048 | 0.917 | 0.876 | 0.933 | 0.909 |
| 4096 | 0.902 | 0.865 | 0.920 | 0.896 |

test set remains in the same domain as the training data but contains forgeries generated by different techniques. This setting evaluates the model's ability to generalize across unseen forgery types. Our method achieves the highest performance by a clear margin in both average and per-method AUC, indicating that it learns more intrinsic, manipulation-agnostic forensic representations.

### 5.3. Ablation Study and Analysis

**Ablation of Key Components.** To systematically verify the contribution of each proposed module, we performed a progressive ablation study on the DFDC and DFDCP datasets, with detailed results summarized in Table 3. The

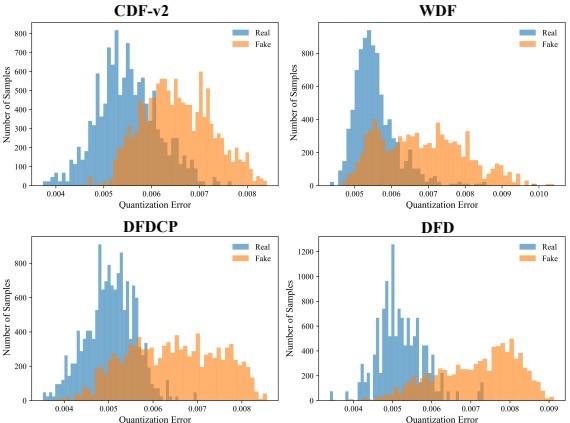

*Figure 6.* **Distribution of quantization errors over the codebook.** We compare the histograms of quantization errors for real and fake samples across different datasets. The significant rightward shift of the fake distributions demonstrates that the discretized anchors effectively reject artifacts, thereby producing informative residual signals for forgery detection.

baseline, *i.e.*, Effort (Yan et al., 2025a), yields an average AUC of 0.876 (see 1th line). Introducing the discrete real codebook elevates the AUC to 0.890 (see 2nd line). Then, the inclusion of the masked ViT restores spatial coherence among tokens, improving the AUC to 0.894 (see 3rd line), while the adaptive tangent space projection (TSP) further refines the detection boundary to 0.900 (see 4th line) by reintroducing locally controllable continuity. Finally, we analyze the impact of the fusion strategy. Note that for the intermediate variants, where the fusion module is absent, the final scores are derived by naively averaging the normalized classifier output logits (see 2nd line) or reconstruction residuals (see 3rd-4th line). By replacing this static averaging with our proposed hypernetwork-based fusion module raises the performance to 0.906 (see 5th line). Overall, the code-

Real Image          Fake Image

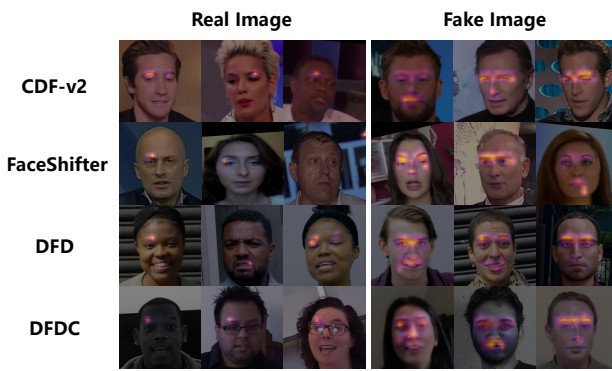

Figure 7. **Spatial visualization of normal residual heatmaps.** We compare the normal residual maps of real and fake samples on CDF-v2 (Yuezun Li & Lyu, 2020), FSh (Li et al., 2020a), DFD (DFD., 2020), DFDC (detection challenge., 2020). The discretized latent modeling yields low-energy responses on real data (left) but produces salient activation peaks on fake data (right).

book gives the biggest boost, while the other parts provide complementary refinements for better robustness.

**Effect of Discrete Real Codebook.** The codebook size $K$ governs the granularity of the discretized latent space. It controls the trade-off between how rich the real features are and how faithfully the information is preserved. We investigate this impact by varying $K$ from 128 to 2048, as shown in Table 4. For a small codebook ($K = 128, 256$) size, the latent manifold space is too sparse to capture the diversity of details, leading to sub-optimal performance due to high reconstruction error even on real data. The performance peaks at $K = 1024$. At this granularity, the codebook provides sufficient capacity to real anchor legitimate variations while keeping a tight bottleneck to reject artifacts.

To further validate the effectiveness of the discrete real codebook, We visualize the distributions of quantization errors for real and manipulated samples across four datasets in Figure 6. Here, the quantization error is defined as the L2 distance between the patch token and its nearest codebook vector. For each image, we compute the average quantization errors over all patch tokens. As hypothesized, real samples exhibit quantization errors tightly concentrated in the lower range, whereas fake samples spread toward higher values. This distributional shift suggests that forgery patterns struggle to find suitable real anchors.

To vividly illustrate the usefulness of proposed reconstruction residuals, we visualize the spatial distribution of normal residuals as heatmaps in Figure 7. For real data, the heatmaps remain largely quiescent with minimal activation. Conversely, for fake data, we observe distinct high-energy activation patterns. Notably, these residual peaks are concentrated around critical facial regions that are most susceptible to generative artifacts.

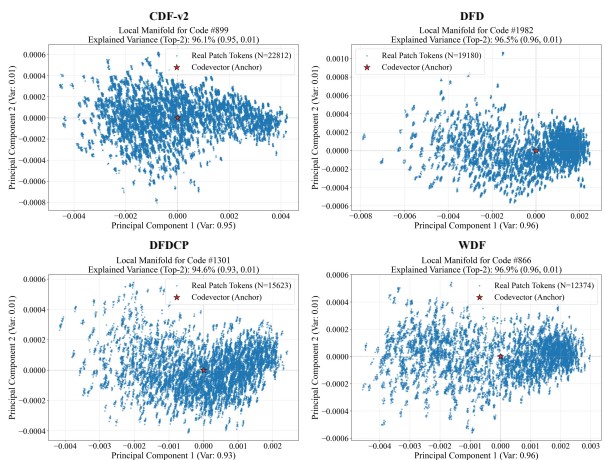

Figure 8. **PCA visualization of local codebook clusters.** We visualize the spatial distribution of real patch tokens (blue dots) assigned to four distinct codevectors (red stars). For each cluster, we report the cumulative explained variance ratio of the top-2 principal components. The high variance coverage indicates that local features lie on a low-dimensional subspace, empirically supporting our linear tangent space projection design.

**Effect of Adaptive Tangent Space Projection.** We introduced the adaptive tangent space projection (TSP) based on the assumption that real patch features assigned to a specific code vector lying on a local low-dimensional linear subspace. To empirically verify this hypothesis, we analyze the geometric structure of the feature clusters using principal component analysis (PCA) on datasets. Note that we employ PCA instead of non-linear manifold visualization techniques like t-SNE, because our TSP models the tangent space as a linear projection ($U_k U_k^T$). Figure 8 shows the distribution of real patch tokens assigned to four randomly selected code vectors without TSP refinement. The clusters form a flattened structure, with the top two principal components explaining over 95% of the variance, indicating that valid real variations (*e.g.*, lighting, pose) lie in a low-rank subspace and thereby justifying the TSP module.

**Robustness under Input Perturbations** We investigate whether our discretized manifold modeling retains its efficacy under input distortions. Adopting the configuration from (Jiang et al., 2020), we introduce four categories of perturbations—Gaussian Blur, JPEG Compression, Color Contrast, and Block-wise noise—each with five varying degrees of severity. Comparative analysis with leading detectors (CLIP (Radford et al., 2021), Forensics Adapter (Cui et al., 2025), SBI (Shiohara & Yamasaki, 2022), and LSDA (Yan et al., 2024)) is presented in Figure 9. The results indicate that our method consistently maintains a high video-level AUC even under severe corruption. We attribute this robustness to the discrete codebook mechanism, which effectively filters out pixel-level high-frequency perturbations while preserving the semantic integrity of the real information.

*Table 5.* **Computational overhead of each model component (Batch size = 32).** We report parameters, GPU memory, and inference latency as modules are progressively added. The full model incurs only ~14% extra latency over the baseline, as the added modules are lightweight and inference is dominated by the frozen CLIP backbone.

| Components | Total Params | Learnable Params | GPU Memory (Occupation) | Inference Time (per 100 batches) |
|---|---|---|---|---|
| Effort (Baseline, Stage 1) | 303.38M | 0.19M | ~7500MB | 16.24s |
| + Codebook ($K = 1024$, Stage 1) | 304.43M | 1.24M (+1.05M) | ~7596MB (+96MB) | 16.59s |
| + Masked ViT (Stage 2) | 330.23M | 27.04M (+25.8M) | ~8273MB (+677MB) | 17.53s |
| + TSP & Fusion (Stage 2) | 348.53M | 45.34M (+18.3M) | ~8893MB (+620MB) | 18.22s |

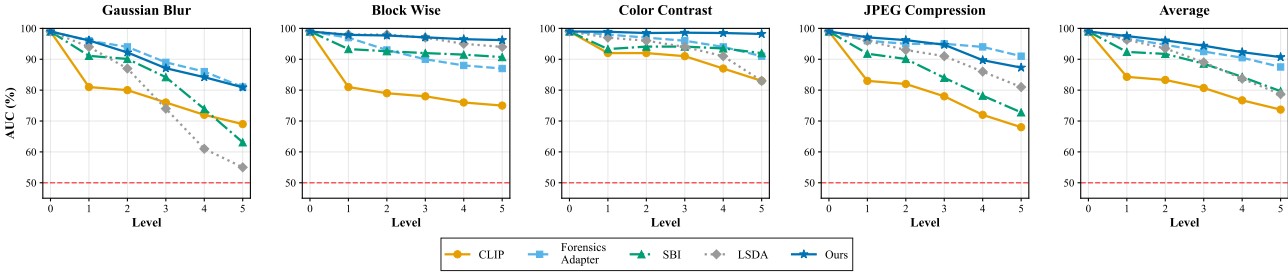

*Figure 9.* **Robustness analysis to unseen perturbations.** Our method is compared with CLIP (Radford et al., 2021), Forensics Adapter (Cui et al., 2025), SBI (Shiohara & Yamasaki, 2022), and LSDA (Yan et al., 2024) across five intensity levels of four types of perturbations, with results reported in terms of video-level AUC.

## 5.4. Component-Wise Computational Overhead

To clarify the precise computational trade-offs introduced by each stage of our framework, we benchmark the total parameters, learnable parameters, GPU memory occupation, and inference latency under identical hardware conditions. All metrics are measured with a fixed batch size of 32. As reported in Table 5, while Stage 2 introduces additional learnable parameters, the inference overhead remains modest: the latency increases from 16.24s to 18.22s per 100 batches (an increase of approximately 14%). This efficiency stems from the fact that our added modules involve only lightweight operations (such as codebook lookup and low-rank projections), leaving the overall inference dominated by the frozen CLIP backbone. In practice, the Stage 1 codebook-only variant offers a low-cost alternative, while the full model targets accuracy-oriented settings.

## 6. Conclusion

In this work, we propose a novel framework for real-centric deepfake detection task. By integrating a vector-quantized codebook into the CLIP visual latent space, we impose a strict information bottleneck that effectively prevents the interpolation of forgery traces. At the same time, our proposed adaptive tangent space projection mechanism carefully balances discrete rigidity and continuous variability, allowing for legitimate intra-class diversity. After that, we obtain discriminative reconstruction residuals that can be directly used for forgery detection. Extensive experiments show

that our method achieves better performances than state-of-the-art approaches. A current limitation is that we rely on a fixed-capacity codebook, which may fail to fully represent extreme facial poses or severe occlusions that deviate significantly from the learned code vectors. Future work will explore dynamic codebook expansion and multi-scale discretization to further improve robustness.

## Impact Statement

This work contributes to the field of media forensics by proposing a robust framework for identifying forgery face images. By enhancing detection capabilities, it supports the integrity of digital information and aids in curbing the spread of disinformation. However, we acknowledge the potential dual-use risk where our insights into manifold discretization could be exploited to refine generative models. To mitigate this, we commit to responsible model release protocols and urge the community to leverage these findings primarily for defensive advancement.

## Acknowledgment

This work was supported by the National Natural Science Foundation of China (Nos. 62302297, 72192821, 62472282, 62272447, and 62472285), the Fundamental Research Funds for the Central Universities (No. YG2023QNA35), and the YuCaiKe Project (No. 231111310300).

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

## A. Theoretical Analysis on Generalization and Interpolation Rejection

To reinforce the core claims of our real-centric paradigm, we present a unified theoretical analysis demonstrating how our discrete-continuous decomposition rejects both interpolation-based forgeries and off-manifold generative artifacts.

### A.1. Mathematical Setup

Our method operates at the patch level: each face image is divided into $M$ patches, and each patch is encoded into a token $x \in \mathbb{R}^D$ ($D = 1024$ from CLIP ViT-L). Let $\mathcal{C} = \{c_k\}_{k=1}^K$ be the discrete codebook, $\mathcal{T}_k$ the $d$-dimensional local tangent space at $c_k$ ($d = 16$ in our configurations), and $\mathcal{P}_k^\perp$ the orthogonal normal residual projection operator ($\mathcal{P}_k^\perp = I - \mathcal{T}_k \mathcal{T}_k^\top$). Let $\mathcal{D}_{\text{real}}$ and $\mathcal{D}_{\text{fake}}$ denote the real and fake patch-token distributions, respectively.

### A.2. Why Discrete Priors Reject Interpolations While Continuous Priors Fail

**Insight 1: The Convexity Failure of Continuous Priors.** Suppose a continuous model (*e.g.*, VAE or GMM) imposes a standard Gaussian prior $p_{cont}(z) = \mathcal{N}(0, \mathbf{I})$. The anomaly score is typically proportional to the negative log-likelihood: $S(z) \propto -\log p_{cont}(z) \propto \|z\|_2^2$.

Based on the well-established concentration of measure phenomenon in high-dimensional spaces, the density of a $D$-dimensional standard Gaussian distribution concentrates almost entirely within a narrow hyperspherical shell. In practice, since the pre-trained CLIP features standardly undergo $L_2$ normalization, the empirical distributions of authentic tokens are geometrically constrained to the identical manifold shell. Consequently, the continuous latent representations of authentic codes are strictly "typical" samples that preclude any vectors from collapsing near the origin, satisfying $\|z_i\|_2^2 \approx d$.

By the strict convexity of the squared Euclidean norm and applying Jensen's Inequality, we derive:

$$\|z_{fake}\|_2^2 = \left\| \sum_{i=1}^N \alpha_i z_i \right\|_2^2 < \sum_{i=1}^N \alpha_i \|z_i\|_2^2 \approx \sum_{i=1}^N \alpha_i d = d. \tag{13}$$

Since the continuous log-likelihood is proportional to $-\|z\|_2^2$, a smaller norm implies a higher probability. This theoretically motivates that continuous priors create a convex "safe zone," inadvertently assigning higher likelihoods to interpolated artifacts than to real samples.

**Insight 2: The Void Property of Discrete Priors.** Conversely, suppose the real discrete modeling is formulated as a mixture of Dirac distributions over the vector quantized codebook $\mathcal{C} = \{c_k\}_{k=1}^K$. Via vector quantization, each real token $x_i$ is mapped to its nearest code vector. Since $z_{fake}$ is a convex combination of distinct code anchors, it constitutes a strict interior point of the segment connecting them.

Because the codebook $\mathcal{C}$ is a finite discrete set, an interior point cannot coincide with any vertex anchors, implying that the distance to the nearest code vector is strictly positive: $\min_k \|z_{fake} - c_k\|_2^2 = \epsilon > 0$. Unlike continuous models that assign finite probabilities to interpolations, the discrete model pushes the anomaly score to infinity as the distribution tightens around the discrete anchors. Thus, $z_{fake}$ inevitably falls into the probability "void" between grid points, ensuring robust rejection of interpolation artifacts.

### A.3. Case 1: Interpolation-Based Forgeries (Face-Swapping / Reenactment)

Forgeries produced by smooth transformations (blending, warping, identity-conditioned generation) can be locally approximated as convex combinations of nearby real patch tokens in feature space.

**Intuition.** Let $x_1, \ldots, x_n \in \mathbb{R}^D$ be real patch tokens quantized to code vectors $c_{(i)} \in \mathcal{C}$. Consider a fake token formed as a non-trivial convex combination $x_{fake} = \sum_{i=1}^n \lambda_i x_i$ with $\lambda_i \geq 0$, $\sum_i \lambda_i = 1$, and at least two distinct anchors among $\{c_{(i)}\}$. By the commitment loss, $x_i \approx c_{(i)}$, so $x_{fake} \approx \bar{c} \triangleq \sum_i \lambda_i c_{(i)}$, which lies in the convex hull $\text{Conv}(c_{(1)}, \ldots, c_{(n)})$. Since $\mathcal{C}$ is a finite discrete set, any non-degenerate interior point of this hull cannot coincide with a codebook element. Letting $c_{fake} = \arg\min_{c_k \in \mathcal{C}} \|x_{fake} - c_k\|$, we have

$$\|x_{fake} - c_{fake}\| \geq \min_{c_k \in \mathcal{C}} \|\bar{c} - c_k\| > 0, \tag{14}$$

which recovers the $\frac{1}{2}\Delta_{\min}$ bound when $n = 2$. The anomaly score $\mathcal{A}(x_{\text{fake}}) = \|\mathcal{P}_k^{\perp}(x_{\text{fake}} - c_{\text{fake}})\|$ vanishes only if the residual lies entirely within the $d$-dimensional tangent space $\mathcal{T}_k$ $(d \ll D)$. Since the residual direction is governed by the discrete codebook layout and shares no prior alignment with $\mathcal{T}_k$, such an event has Lebesgue measure zero. Therefore $\mathcal{A}(x_{\text{fake}}) > 0$, explaining why convex mixtures of real tokens, a common behavior of generative models, are naturally exposed by our residual-based detector.

### A.4. Case 2: Off-Manifold Forgeries (GAN / Diffusion Synthesis)

Generative models that synthesize forged images from holistic spaces do not strictly simulate local real combinations, but instead introduce structural deviations.

**Intuition.** For a given fake patch token $x_{\text{fake}}$, let $x_{\text{fake}} = x_{\text{real}} + \delta$, where $x_{\text{real}}$ is the corresponding oracle real patch token on the natural manifold and $\delta$ represents the generative artifact perturbation. Expanding the normal residual via the reverse triangle inequality:

$$\|\mathcal{P}_k^{\perp}(x_{\text{fake}} - c_k)\| = \|\mathcal{P}_k^{\perp}(x_{\text{real}} + \delta - c_k)\| \tag{15}$$

$$= \|\mathcal{P}_k^{\perp}\delta + \mathcal{P}_k^{\perp}(x_{\text{real}} - c_k)\| \tag{16}$$

$$\geq \|\mathcal{P}_k^{\perp}\delta\| - \|\mathcal{P}_k^{\perp}(x_{\text{real}} - c_k)\|. \tag{17}$$

The Tangent Space Projection (TSP) loss explicitly minimizes the term $\|\mathcal{P}_k^{\perp}(x_{\text{real}} - c_k)\| \leq \epsilon$ for natural real samples, establishing the lower bound

$$\|\mathcal{P}_k^{\perp}(x_{\text{fake}} - c_k)\| \geq \|\mathcal{P}_k^{\perp}\delta\| - \epsilon. \tag{18}$$

Since the training process minimizes the codebook commitment and tangent space loss on real data, $\epsilon \to 0$, yielding $\|\mathcal{P}_k^{\perp}(x_{\text{fake}} - c_k)\| \gtrsim \|\mathcal{P}_k^{\perp}\delta\|$.

**Justification for $\|\mathcal{P}_k^{\perp}\delta\| > 0$.** We justify why the artifact component $\delta$ cannot hide inside the tangent space $\mathcal{T}_k$ through two complementary properties:

1. **Dimensionality Constraints:** The tangent space $\mathcal{T}_k$ has a strictly constrained dimension ($d = 16$). For the artifact vector $\delta$ to yield a zero normal residual ($\|\mathcal{P}_k^{\perp}\delta\| = 0$), it must be completely embedded within this 16-dimensional subspace of the 1024-dimensional embedding space. For any arbitrary distribution on $\mathbb{R}^D$ absolutely continuous with respect to the Lebesgue measure, the probability of falling into a lower-dimensional subspace is exactly zero ($P(\delta \in \mathcal{T}_k) = 0$). Statistically, the expected energy fraction outside the subspace is $1 - (16/1024) = 98.43\%$.

2. **Structural Independence:** The principal axes of $\mathcal{T}_k$ are learned exclusively from the local covariance $\Sigma_{\text{real}}$ of authentic real samples. Because generative artifacts ($\delta$) originate from specific upsampling layers, architectural grid biases, or diffusion inversion paths independent of the natural human face manifold, their mutual information $\mathcal{I}(\delta; \mathcal{T}_k) \approx 0$. Hence, the artifact vector $\delta$ is statistically unaligned with the eigenvectors of $\Sigma_{\text{real}}$, confirming $\|\mathcal{P}_k^{\perp}\delta\| > 0$.

**Summary.** Our framework establishes two complementary lines of defense: interpolation-based deepfakes are confined within the discrete "voids" between codebook anchors, while out-of-domain GAN/Diffusion forgeries are rejected via their projection onto the orthogonal complement of the local tangent spaces. This dual mechanism jointly ensures the generalization of our framework.

## B. Empirical Verification of Continuous Prior Convexity

To empirically validate our theoretical claim regarding the "Convexity Failure of Continuous Priors" (A.2 Insight 1), we have conducted a likelihood analysis using real face data. The objective is to demonstrate that continuous probabilistic models (*e.g.*, GMMs over deep features) tend to assign higher likelihood probabilities to interpolated "fake" samples than to the authentic samples themselves, due to the concentration of measure phenomenon.

**Experimental Setup.** We employed a pre-trained CLIP-ViT/L vision encoder to extract latent feature vectors from the authentic subset of the dataset. A Gaussian Mixture Model (GMM) with a diagonal covariance matrix was then fitted to these features to approximate the continuous manifold of real faces. We randomly sampled pairs of authentic feature vectors

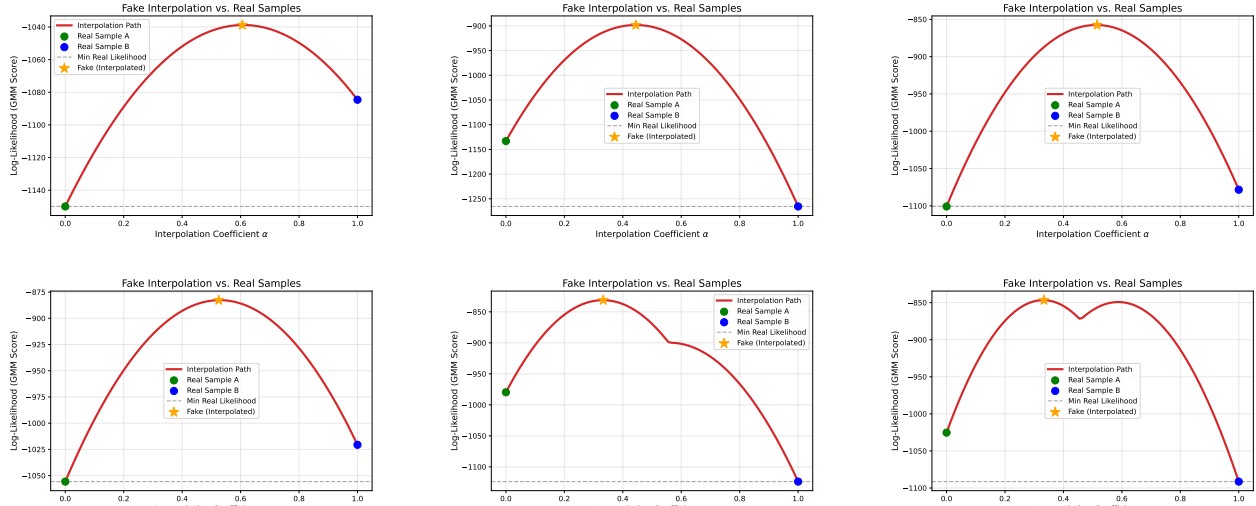

*Figure 10.* **Empirical verification of the likelihood in continuous modeling on FF++ dataset.** We perform linear interpolation between pairs of randomly sampled authentic face features extracted by CLIP. The red curves illustrate the log-likelihood scores estimated by a GMM trained exclusively on real data. In all cases, the interpolated "fake" points (near $\alpha = 0.5$) exhibit significantly higher likelihoods than the authentic endpoints ($z_a, z_b$), creating an inverted U-shaped trajectory. This empirically validates that continuous manifolds suffer from the concentration of measure, failing to reject—and even favoring—interpolated anomalies.

$(z_a, z_b)$ and performed linear interpolation $z_{interp} = \alpha z_a + (1 - \alpha)z_b$ with $\alpha \in [0, 1]$. We then evaluated the log-likelihood scores of the interpolation path using the trained GMM.

**Observations.** Figure 10 illustrates the log-likelihood curves for 6 randomly selected pairs of authentic samples. In all cases, we observe a distinct inverted U-shaped curve. Specifically, the likelihood score rises as the interpolation coefficient $\alpha$ approaches 0.5 and peaks at the center, significantly surpassing the scores of the two genuine endpoints.Conclusion. These results empirically confirm that continuous priors create a high-probability "safe zone" in the latent space between real samples. Consequently, deepfakes that fall within this convex hull (*e.g.*, through smooth interpolation or reconstruction) are not only accepted but are often deemed "more likely" than actual real faces. This fundamental flaw justifies our proposed shift towards discrete modeling, which eliminates this ambiguity by enforcing a hard information bottleneck.

## C. Additional Implementation Details

To supplement the architectural specifics omitted in the main text due to space constraints, we provide the comprehensive configurations of the masked ViT, hypernetwork, and classification head below.

### C.1. Masked ViT Configuration

The masked ViT operates on input patch tokens with an embedding dimension of $1024$ and a total of $256$ patch tokens. During training, we employ a high masking ratio of $0.75$, where the masked patches are systematically replaced by a single learnable mask token. The encoder architecture is lightweight, consisting of $2$ transformer blocks, each configured with $8$ attention heads.

### C.2. Hypernetwork & Classification Head

The hypernetwork is designed as a two-layer multi-Layer perceptron layers (MLP) that dynamically conditions the classification. It takes the $1024$-dimensional `[CLS]` token from the backbone as input. The network structure is formalized as: $\text{Linear}(1024, 512) \rightarrow \text{ReLU} \rightarrow \text{Linear}(512, 2048)$. The resulting $2048$-dimensional output vector is subsequently reshaped to dynamically generate the classifier weights of size $1024 \times 2$ for binary prediction, which are accompanied by a standard learnable bias vector of size 2.

## D. Efficiency Comparison Against Existing Methods

We further evaluate our method against existing state-of-the-art architectures in terms of parameterized efficiency and operational speed. As shown in Table 6, despite our framework utilizing a two-stage training scheme that increases training costs, its inference runtime remains highly competitive. Our full model achieves an inference time of 18.22s per 100 batches, which is remarkably close to the baseline Effort (16.24s) despite a larger learnable parameter count (45.34M vs. 0.19M), and substantially faster than Forensics Adapter (32.55s) and FatFormer (40.22s).

*Table 6.* **Computational overhead against existing methods (Batch size = 32).** We report the total/learnable parameters and inference time of representative methods. Despite our two-stage training scheme, the inference runtime remains highly competitive.

| Methods | Total Params | Learnable Params | Inference Time (per 100 batches) |
|---|---|---|---|
| LSDA (Yan et al., 2024) | 133M | 133M | 15.48s |
| Forensics Adapter (Cui et al., 2025) | 435.16M | 7.55M | 32.55s |
| FatFormer (Liu et al., 2024) | 577.25M | 94.53M | 40.22s |
| Effort (Yan et al., 2025a) | 303.38M | 0.19M | 16.24s |
| Ours | 348.53M | 45.34M | 18.22s |

## E. Generalizability to General AIGC Benchmarks

While our framework is intrinsically tailored for facial forgery detection, with a fixed codebook capacity ($K = 1024$) aligned to facial structural priors, we further assess its generalizability beyond the facial domain on the GenImage benchmark. As shown in Table 7, with only minor hyperparameter adjustments for open-world semantics, our method attains competitive results across diverse diffusion- and GAN-based generators, reaching an overall mean accuracy of 92.3% and outperforming the strong CLIP-based baseline Effort by 1.2%. These results highlight the robustness and scalability of our approach in general AIGC detection scenarios.

*Table 7.* **Evaluation results (Acc %) on the GenImage dataset.** We train the model on the training set of Stable Diffusion v1.4 (SDv1.4) and evaluate it on the test sets of other subsets.

| Methods | SDv1.5 | Wukong | VQDM | ADM | BigGAN | GLIDE | Midjourney | SDv1.4 | mAcc |
|---|---|---|---|---|---|---|---|---|---|
| UnivFD (Ojha et al., 2023b) | 96.1 | 94.5 | 67.8 | 58.1 | 57.7 | 73.4 | 91.5 | 96.4 | 79.5 |
| NPR (Tan et al., 2024b) | 97.9 | 96.9 | 84.1 | 76.9 | 84.2 | 89.8 | 81.0 | 98.2 | 88.6 |
| FreqNet (Tan et al., 2024a) | 98.6 | 97.3 | 76.8 | 66.8 | 81.4 | 86.5 | 89.6 | 98.8 | 86.8 |
| FatFormer (Liu et al., 2024) | **99.9** | **99.9** | **98.8** | 75.9 | 55.8 | 88.0 | **92.7** | **100.0** | 88.9 |
| DRCT (Chen et al., 2024) | 94.4 | 94.7 | 90.0 | **79.4** | 81.7 | 89.2 | 91.5 | 95.0 | 89.5 |
| Effort (Yan et al., 2025a) | 99.8 | 97.4 | 91.7 | 78.7 | 77.6 | 93.3 | 82.4 | 99.8 | 91.1 |
| Ours | 99.7 | 99.1 | 94.6 | 79.1 | **88.4** | **98.7** | 79.5 | 99.8 | **92.3** |

## F. Algorithmic Workflows and Framework Execution

To provide a concrete blueprint for hardware implementation and reproduction, this section details the operational pipelines of our framework. We present the decoupled two-stage training paradigm in Algorithm 1 and the complete test-time inference pipeline in Algorithm 2.

---

**Algorithm 1** Training Scheme of the Proposed Framework

---

**Require:** Training dataset $\mathcal{D}$, CLIP encoder $E$, Codebook $\mathcal{C}$, Masked ViT $M$, HyperNetwork $\Psi$, TSP Bases $\{U_k\}$.
 1: **Stage 1: Discrete Codebook Learning**
 2: Initialize $E$ with LoRA or Effort, randomly init $\mathcal{C}$.
 3: **while** not converged **do**
 4:     Sample batch of authentic images $\{x_{real}\} \sim \mathcal{D}_{real}$.
 5:     Extract features: $Z, z_{cls} = E(x_{real})$.
 6:     Quantize patch tokens: $Z_q = \text{Quantize}(Z, \mathcal{C})$.
 7:     Compute Loss ($\mathcal{L}_{\text{COM}} + \mathcal{L}_{\text{DIS}}$).
 8:     Update $E$ (LoRA or Effort params) and $\mathcal{C}$.
 9: **end while**
10: **End Stage 1**. Freeze $E$ and $\mathcal{C}$.
11: **Stage 2: Adaptive TSP & HyperNetwork Optimization**
12: **while** not converged **do**
13:     Sample batch $\{x, y\} \sim \mathcal{D}$ (Real and Fake).
14:     Get frozen quantized tokens: $Z_q, z_{cls} = E(x)$.
15:     Apply random masking: $\tilde{Z}_q = \text{Mask}(Z_q)$.
16:     Reconstruct features: $H = M(\tilde{Z}_q)$.
17:     **Parallel Stream Computing:**
18:     **for** each patch token $h_i$ assigned to code vector $e_k$ **do**
19:         Compute Normal Residual: $r_i = ||(I - U_k^i U_k^{i^T})(h_i - z_i)||_2$.
20:     **end for**
21:     Aggregate geometric residuals: $\mathcal{R} = [r_1, \ldots, r_L]$.
22:     Generate adaptive params: $[W, b^\star] = \Psi(z_{cls})$.
23:     Predict score: $\hat{y} = \sigma(F(\mathcal{R}; W, b^\star))$.
24:     Compute Loss ($\mathcal{L}_{\text{MIM}} + \mathcal{L}_{\text{TSP}} + \mathcal{L}_{\text{CLS}}$).
25:     Update $M$, $\{U_k\}$, and $\Psi$.
26: **end while**

---

**Algorithm 2** Inference Stage

---

**Require:** Test image $x$, Trained components ($E, \mathcal{C}, M, \{U_k\}, \Psi$).
**Ensure:** $p(x)$.
 1: **Feature Extraction:**
 2: $Z, z_{cls} = E(x)$. {Extract via frozen CLIP}
 3: $Z_q = \text{Quantize}(Z, \mathcal{C})$. {Map to discrete scaffold}
 4: **Geometric Refinement (No Masking):**
 5: $H = M(Z_q)$. {Process full context}
 6: **for** $i = 1$ to $L$ **do**
 7:     Identify code vector index $k$ for patch $i$.
 8:     Calculate Tangent Space Residual (Normal Component):
 9:     $r_i = ||(I - U_k^i U_k^{i^T})(h_i - z_i)||_2$.
10: **end for**
11: Formulate spatial residual profile $\mathcal{R} = [r_1, \ldots, r_L]$.
12: **Semantic Adaptation & Decision:**
13: Generate instance-aware weights: $[W, b^\star] = \Psi(z_{cls})$.
14: Compute final probability: $p(x) = \sigma(F(\mathcal{R}; W, b^\star))$.

---

