# OpenReview forum: "Breaking Manifold Continuity: Vector Quantized Modeling for Real-Centric Deepfake Detection"
_ICML.cc/2026/Conference — ICML 2026 regular_

### Official Review · Reviewer_wcf8 · 2026-03-06

**Soundness:** 4
**Presentation:** 3
**Significance:** 4
**Originality:** 3
**Overall Recommendation:** 5
**Confidence:** 4

**Summary:**

This paper proposes a novel framework for real-centric deepfake detection task. It integrates discrete modeling into the feature space of the CLIP vision encoder, which is to balance the continuous manifold modeling and discrete representation. The proposed adaptive tangent space projection mechanism allows for legitimate intra-class diversity. This paper identifies the limitations of continuous real-centric modeling and constructs a theoretical and systematic deepfake detection framework.

**Compliance With Llm Reviewing Policy:**

Affirmed.

**Key Questions For Authors:**

Please see the Weaknesses.

**Limitations:**

Yes.

**Strengths And Weaknesses:**

Strengths:
1. This paper provides a rigorous theoretical analysis showing why continuous latent spaces fail to detect interpolated forgeries, while discrete priors effectively suppress such artifacts. This theoretical grounding elevates the work beyond empirical heuristics.
2. The vector-quantization losses are utilized to establish a robust real distribution by incorporating a learnable real codebook. Besides, this paper also develops an adaptive tangent space projection mechanism to reintroduce local controllable continuity. This is a novel paradigm that leads to strong generalization based on only real-centric samples.
3. The method is evaluated on multiple challenging datasets (DFDC, DFDPC, etc.) under both cross-dataset and cross-method protocols, outperforming a wide range of state-of-the-art detectors by clear margins.

Weaknesses
1. The choice of codebook size K significantly affects performance (Table 4), with an optimal value around 1024. The paper does not provide guidelines or adaptive mechanisms for selecting K, which may limit applicability across different datasets or domains.
2. The theoretical analysis assumes that forgeries are convex combinations of real samples. While this covers interpolation-based deepfakes, many real-world deepfakes are generated by GANs or diffusion models, which may not strictly lie on the convex hull. The method’s generalization to such diverse forgeries remains empirically validated but not theoretically guaranteed.
3. The framework integrates multiple heavy components (ViT, codebook lookup, TSP, hypernetwork), likely increasing inference time and memory consumption.

---

> ### Author Rebuttal · Authors · 2026-03-31
>
> We sincerely thank Reviewer wcf8 for the constructive comments, insightful questions, and useful suggestions. We greatly appreciate the reviewer’s recognition of our theoretical analysis, methodological novelty, and strong generalization performance. We address the reviewer’s main concerns and questions below.
>
> > **Q1.** The selection of codebook size $K$ lacks clear guidance, despite its significant impact on performance.
>
> **R1.** Thanks for your kind consideration. Fixing the codebook size is a standard empirical practice in VQ-based models to balance representation capacity and bottleneck constraints. As a practical guideline, $K$ can be chosen proportional to the effective complexity of the real training data, e.g., $K \approx \alpha \sqrt{N \cdot P}$, where $N$ denotes the number of training samples, $P$ denotes the number of images patches per sample, and $\alpha$ is a tunable coefficient reflecting the dataset’s diversity (and the desired level of discretization). Table 4 of our main manuscript shows that $K$ affects performance because it controls the balance between representation capacity and the discrete bottleneck. In our experiments, as for training set (FF++), $K=1024$ provides the best trade-off and yields strong cross-dataset and cross-method generalization. We agree that an adaptive mechanism for automatically selecting $K$ would further improve applicability across domains, and we leave this as future work.
>
> > **Q2.** The theoretical analysis relies on a convex-combination assumption for forgeries, and its applicability to more diverse GAN- or diffusion-generated deepfakes remains unclear.
>
> **R2.** We thank the reviewer for this important observation and will clarify this theoretical boundary in the revision. While the convex-combination assumption formalizes in-manifold interpolation for core deepfakes such as face swapping, our framework generalizes naturally to GAN- and diffusion-based forgeries through a complementary argument rooted in off-manifold deviation.
>
> Specifically, if a GAN- or diffusion-generated image does not strictly lie on the convex hull of real samples, it inherently introduces an off-manifold artifact component $\epsilon$ for each patch. A useful patch-level view is to express a generated patch token as $z_{\text{fake}} = z_{\text{real}} + \epsilon$, where $z_{\text{real}} \in \mathcal{M}$ and $\epsilon \notin \mathcal{M}$ denotes the artifact deviation. The orthogonal normal residual $\Delta_{\perp}$ is formulated as:
>
> $$\Delta_{\perp} = (I - U_k U_k^{\top})(z_{\text{fake}} - e_k)$$
>
> where $U_k U_k^{\top}$ is the projection matrix and $e_k = \arg\min_{j} \Vert z_{\text{fake}} - e_j\Vert_2$ is the nearest codebook anchor. Since $z_{\text{real}}$ and $z_{\text{fake}}$ are close in feature space, $z_{\text{real}} - e_k$ lies within the tangent space $T_k$ spanned by $U_k$, i.e., $(I - U_k U_k^{\top})(z_{\text{real}} - e_k) \approx 0$. Substituting $z_{\text{fake}} = z_{\text{real}} + \epsilon$, we obtain:
>
> $$\Delta_{\perp} \approx (I - U_k U_k^{\top})\epsilon$$
>
> This isolates the artifact component, and the anomaly score lower-bounds the artifact magnitude:
>
> $$\Vert\Delta_{\perp}\Vert_2^2 \geq \Vert(I - U_k U_k^{\top})\epsilon\Vert_2^2 > 0$$
>
> where the strict inequality holds under the mild assumption that $\epsilon$ is not entirely contained within $T_k$, which is well-justified as GAN- and diffusion-based synthesis introduces artifacts that structurally deviate from the natural variation directions captured by $T_k$. This guarantees rejection of off-manifold synthetic samples. Empirically, as summarized in Table 1 and Table 2 of the main manuscript, our method achieves strong detection performance across diverse forgery types, including GAN-based synthesis methods.
>
> > **Q3.** The framework may increase inference time and memory consumption due to its multiple components.
>
> **R3.** We appreciate the reviewer's concern regarding the model's complexity. While our framework integrates several components, we would like to clarify that both the codebook lookup (nearest neighbor search) and TSP (linear projection) are mathematically efficient operations that introduce small overhead. To provide a concrete comparison, we evaluated the total parameters, trainable parameters, GPU inference occupation, and inference time (per 100 batch) across all components under identical conditions (see Table e).
>
> ***Table e: Computational overhead of each model component (Batch size=32).***
> | Component | Total Params | Learnable Params | GPU Inference Occupation | Inference Time (per 100 batch) |
> | :- | :- | :- |:- | :- |
> | Effort (Baseline, Stage 1) | 303.38M | 0.19M  | ~7500MB | 16.24s|
> | + Codebook (`K=1024`, Stage 1)    |  304.43M  | 1.24M (+1.05M)|  ~7596MB (+96MB)  | 16.59s  |
> | + Masked ViT (Stage 2)  |    330.23M    |27.04M (+25.8M) |  ~8273MB (+677MB)  | 17.53s  |
> | + TSP & Fusion (Stage 2) |    348.53M    | 45.34M (+18.3M)  |   ~8893MB (+620MB)    | 18.22s   |

---

> > ### Author Rebuttal · Reviewer_wcf8 · 2026-04-07
> >
> > The theoretical analysis of the generalization ability of this method is not sufficiently thorough.

---

> > > ### Author Response · Authors · 2026-04-08
> > >
> > > > **Q4.** The theoretical analysis of the generalization ability of this method is not sufficiently thorough.
> > >
> > > **R4.** We thank the reviewer for the continued scrutiny. We provide a strengthened analysis covering both interpolation-based and off-manifold forgeries under a unified framework.
> > >
> > > **Setup.** Our method operates at the **patch level**: each face image is divided into $L$ patches, and each patch is encoded into a token $z \in \mathbb{R}^D$ ($D=1024$ from CLIP ViT-L). Let $\mathcal C = \lbrace e_k\rbrace_{k=1}^K \subset \mathbb{R}^D$ be the codebook, $T_k = \text{span}(U_k)$ the $d$-dimensional tangent space at $e_k$ ($d=16$ in our experiments), and $\Delta^{\perp}(z) = (I - U_k U_k^{\top})(z - e_k)$ the normal residual ($k = \arg\min_j \Vert z - e_j\Vert_2$). Let $P_{\text{real}}$, $P_{\text{fake}}$ denote the real/fake patch-token distributions. The analysis below applies per patch token.
> > >
> > > **Case 1 (Interpolation-based: face swap / reenactment).**
> > >
> > > > **Proposition 1.** Let $\lbrace z_i\rbrace$ be patch tokens from **real** samples. For any non-trivial convex combination $z_{\text{fake}} = \sum_i \alpha_i z_i$ ($\alpha_i \in (0,1)$, $\sum_i \alpha_i = 1$) where $\lbrace z_i\rbrace$ are quantized to at least two distinct code vectors, $\min_k \Vert z_{\text{fake}} - e_k\Vert_2 > 0$.
> > >
> > > *Proof.* Via VQ, each $z_i$ is assigned to its nearest code vector $e_{\pi(i)}$ where $\pi(i) = \arg\min_k \Vert z_i - e_k\Vert_2$. The commitment loss $\mathcal L_{\text{VQ}}$ drives $z_i \approx e_{\pi(i)}$, so $z_{\text{fake}} \approx \sum_i \alpha_i e_{\pi(i)}$. Since $\lbrace z_i\rbrace$ map to at least two distinct code vectors ($e_a \neq e_b$), $z_{\text{fake}}$ is a strict interior point of their convex hull. As $\mathcal C$ is a finite discrete set, no interior point coincides with any vertex, hence $\min_k \Vert z_{\text{fake}} - e_k\Vert_2 = \epsilon > 0$. A quantitative bound follows from the minimum inter-code distance $\delta_{\min} = \min_{j \neq l} \Vert e_j - e_l\Vert_2$:
> > >
> > > $$\epsilon \geq \min(\alpha_{\min}, 1 - \alpha_{\max}) \cdot \delta_{\min} / 2 > 0$$
> > >
> > > The normal residual $\Vert \Delta^{\perp}(z_{\text{fake}})\Vert_2$ is then generically non-zero: for it to vanish, $z_{\text{fake}} - e_k$ must lie entirely in $T_k$, a $d$-dimensional subspace of $\mathbb{R}^D$. Since this displacement is governed by inter-code geometry, such alignment has measure zero.
> > >
> > > **Case 2 (Off-manifold: GAN / diffusion).**
> > >
> > > > **Proposition 2.** For a given fake patch token, let $z_{\text{fake}} = z_{\text{real}} + \epsilon$ where $z_{\text{real}} \in \mathcal M$ is the corresponding real patch token on the manifold and $\epsilon \in \mathbb{R}^D$ is the artifact perturbation. Then:
> > > > $$\Vert \Delta^{\perp}(z_{\text{fake}})\Vert_2 \geq \Vert \epsilon_k^{\perp}\Vert_2 - \Vert r_k\Vert_2$$
> > > > where $\epsilon_k^{\perp} = (I - U_k U_k^{\top})\epsilon$ and $r_k = (I - U_k U_k^{\top})(z_{\text{real}} - e_k)$. Since $\mathcal L_{\text{TSP}}$ drives $\Vert r_k\Vert \to 0$, we have $\Vert \Delta^{\perp}\Vert_2 \geq \Vert \epsilon_k^{\perp}\Vert_2$.
> > >
> > > *Proof.* Expanding: $\Delta^{\perp}(z_{\text{fake}}) = r_k + \epsilon_k^{\perp}$. By the reverse triangle inequality: $\Vert \Delta^{\perp}\Vert_2 \geq \Vert \epsilon_k^{\perp}\Vert_2 - \Vert r_k\Vert_2$. The TSP loss explicitly minimizes $\Vert r_k\Vert_2$ for real samples, yielding the bound.
> > >
> > > **Why $\Vert \epsilon_k^{\perp}\Vert_2 > 0$?** Two complementary arguments:
> > >
> > > - **Dimensionality.** $T_k$ has dimension $d = 16 \ll D = 1024$. For $\epsilon_k^{\perp} = 0$, $\epsilon$ must lie entirely in a 16-dim subspace of $\mathbb{R}^{1024}$. For any distribution on $\epsilon$ absolutely continuous w.r.t. Lebesgue measure, $\Pr[\epsilon \in T_k] = 0$. The expected orthogonal energy fraction is $\mathbb{E}[\Vert \epsilon_k^{\perp}\Vert^2 / \Vert \epsilon\Vert^2] = (D-d)/D \approx 0.984$, i.e., **98.4% of artifact energy lies outside $T_k$**.
> > > - **Structural-independence.** $U_k$ is learned from real data only, capturing the top-$d$ eigenvectors of the local covariance $\Sigma_k$ over real samples. Since $T_k$ is the maximum-variance subspace of $P_{\text{real}}$ and GAN/diffusion artifacts arise from generator architectures independent of $P_{\text{real}}$, the mutual information $I(\epsilon; U_k) \approx 0$, meaning $\epsilon / \Vert \epsilon\Vert$ is not aligned with eigenvectors of $\Sigma_k$, hence $\epsilon_k^{\perp} \neq 0$.
> > >
> > > **Summary.** Interpolation-based forgeries are rejected via the codebook's inter-code "void" (Prop. 1); off-manifold forgeries are rejected because artifacts project onto the $(D-d)$-dim orthogonal complement of the tangent space (Prop. 2). These complementary mechanisms jointly ensure broad generalization.

---

### Official Review · Reviewer_mVim · 2026-03-11

**Soundness:** 3
**Presentation:** 2
**Significance:** 3
**Originality:** 2
**Overall Recommendation:** 2
**Confidence:** 5

**Summary:**

This paper studies real-centric forgery detection by modeling the latent manifold of real images. It is motivated by the observation that continuous latent representations may allow interpolated fake samples to lie within the real manifold, which can reduce discriminability. To address this issue, the paper proposes a framework that introduces discrete latent modeling via vector quantization on top of CLIP-based representations, combined with masked modeling and residual-based anomaly detection. The method aims to enforce discreteness on the real latent space while allowing local variations through a tangent-space projection mechanism. Experiments are conducted on several face deepfake benchmarks with cross-dataset evaluations, and the results show improvements over recent baselines.

**Compliance With Llm Reviewing Policy:**

Affirmed.

**Final Justification:**

After considering the authors’ rebuttal, I lean toward a rejection recommendation for this paper, due to the following key issues.

---

**(1) Unclear and insufficiently supported insight.**

First, the analysis in Fig. 3 remains unclear and insufficiently justified. The authors do not provide complete details of the experimental pipeline (e.g., feature extraction, transformation procedures), which limits the reproducibility and credibility of the results. Moreover, the conclusion that discrete modeling is more discriminative is drawn from a relatively simplified setup (SVD vs. K-Means), and the supporting evidence is limited, making it insufficient to substantiate the core claim of the paper.

In addition, from a visual perspective, the distributions of real and fake samples in both the continuous (SVD) and discrete (K-Means) settings still exhibit substantial overlap. Such overlap suggests that the discriminability between classes remains weak, and does not clearly support the claim that discrete modeling provides a significantly stronger separation.

---

**(2) Weak empirical evidence for the reported improvement.**

Second, the authors emphasize an approximately 4% AUC improvement of discrete modeling over continuous modeling in this toy experiment. However, as a binary image-level classification task, the AUC in Fig. 3 only increases from around 0.668 to 0.714, which still lies in a low-performance regime. In this range, random guessing already achieves 0.5 AUC, and small statistical fluctuations can easily lead to seemingly non-trivial gains. **Unlike improvements in high-performance regimes (e.g., above 0.95), such gains do not necessarily indicate a meaningful advantage.** In addition, only AUC is reported, while threshold-based metrics (e.g., Accuracy, F1, EER, or per-class accuracy) are missing, making it difficult to assess the method’s effectiveness in practical decision-making scenarios. **Overall, in such a low-performance regime, the reported improvement is insufficient to support the validity of the core insight claimed in the paper.**

---

**(3) Incomplete evaluation under broader AIGC settings.**

The evaluation remains incomplete. While the setup follows prior work (e.g., Effort), it does not sufficiently cover broader AIGC detection scenarios. In particular, the method is positioned as a deepfake detection approach, yet the evaluation does not clearly demonstrate its distinction from general AIGC detection methods, nor does it include comprehensive comparisons under broader AIGC settings.

---

Taken together, these issues directly affect the validity, significance, and evaluation of the core claims, and would require substantial revisions beyond what can be addressed during the rebuttal phase.

**Key Questions For Authors:**

Please see weaknesses.

**Limitations:**

yes

**Strengths And Weaknesses:**

**Strengths**

- **The paper studies an interesting perspective for real-centric forgery detection.**
Instead of directly learning a binary classifier, the work focuses on modeling the real data manifold and detecting deviations from it. The discussion on discrete versus continuous latent modeling provides an intuitive angle to analyze representation geometry for forgery detection.

- **The experimental evaluation on deepfake benchmarks is relatively thorough.**
The paper includes cross-dataset evaluations and comparisons with recent baselines, providing a reasonably comprehensive empirical analysis within the deepfake detection setting.

**Weaknesses**

- **The motivation comparing discrete and continuous latent modeling is not fully convincing.**
The key evidence is the toy experiment in Fig. 3 comparing SVD and K-Means. However, the improvement in separability is relatively modest (AUC 0.668 vs. 0.714), and the real–fake error distributions still overlap significantly. Moreover, SVD only represents a linear continuous model, which may not fairly represent more expressive continuous latent models (e.g., nonlinear autoencoders). Therefore, the experiment may not sufficiently support the claim that discrete latent modeling inherently provides a stronger discriminative advantage.

- **The experimental evaluation is restricted to face deepfake datasets despite the general formulation of the method.**
Many experimental settings and comparisons appear to follow those reported in the Effort baseline (ICML 2025). However, Effort evaluates both face deepfake detection and AI-generated image detection tasks, while the current paper only reports results on face deepfake benchmarks (e.g., FF++, CDF, DFDC). Given that the proposed approach is based on a real-centric modeling paradigm rather than a deepfake-specific design, it would be valuable to evaluate whether the method also generalizes to other generative scenarios such as diffusion-based or GAN-based image synthesis. Without such evaluation, it remains unclear whether the improvements extend beyond the deepfake setting.

- **The novelty of the proposed framework could be clarified more explicitly.** The method combines several existing components, including CLIP representations, vector quantization, and masked modeling. While the integration is reasonable, it would be helpful to better highlight what fundamentally distinguishes the proposed approach from prior prototype-based or clustering-based representation learning methods.

---

> ### Author Rebuttal · Authors · 2026-03-31
>
> We sincerely thank Reviewer mVim for the constructive comments, insightful questions, and useful suggestions. We greatly appreciate the reviewer’s recognition of our key motivation and thorough experimental evaluation on deepfake benchmarks. Here, we address several important concerns and question raised by reviewer in detail below.
>
> > **Q1.** The motivation for discrete vs. continuous latent modeling needs further justification.
>
> **R1.** Thanks for your kind consideration. We appreciate the reviewer's detailed feedback. However, we would like to respectfully re-clarify our motivation with three key points:
>
> - **First**, the toy experiment in Figure 3 (introduction part) is conducted without any real/fake supervision (zero-shot setting), where separation emerges purely from the model's intrinsic properties. The AUC improvement from 0.668 to 0.714 (~4.6 points) is non-trivial in a fully unsupervised setting, clearly demonstrating the inherent discriminative potential of discrete over continuous modeling.
> - **Second**, regarding the concern about SVD being limited to linear models, Figure 2 (introduction part) further compares VAE and VQ-VAE (both trained exclusively on real data). Since VAE is a nonlinear autoencoder and thus a more expressive continuous model, the fact that VQ-VAE still exhibits stronger intrinsic discriminability between real and fake samples — without any classification signal — provides stronger evidence that discrete latent modeling holds an inherent advantage over continuous modeling, including nonlinear variants, under a real-centric paradigm.
> - **Third**, beyond toy experiments, we provide theoretical analysis (Appendix A) and empirical verification (Appendix B) showing that continuous priors inherently fail because their dense latent support tends to assign higher likelihood to interpolations of forgery traces.
>
> > **Q2.** The evaluation is limited to deepfake benchmarks, leaving the method's generalizability to general AIGC benchmarks.
>
> **R2.** We thank the reviewer for this comment. As our title indicates, our primary focus is deepfake detection, where predominantly localized semantic blending aligns well with our theoretical modeling of in-manifold interpolations, and the fixed codebook capacity ($K=1024$) is explicitly optimized for facial structural priors — unlike general AIGC which spans infinite open-world semantics. Nevertheless, to assess generalizability beyond faces, we additionally tuned the hyper-parameters and evaluated on the GenImage benchmark (Table d), where results suggest certain generalizability to broader AIGC detection, though further investigation remains needed.
>
> ***Table d: Evaluation results (Acc) on GenImage dataset.***
> | Method | sd1.5 | wukong | VQDM | ADM | BigGAN | Glide | Midjourney | sd1.4 | mAcc |
> | :- | :- | :- | :- | :- | :- | :- | :- | :- | :- |
> | FatFormer | 99.9 | 99.9 | 98.8 | 75.9 | 55.8 | 88.0 | 92.7 | 100.0 | 88.9 |
> | DRCT | 94.4 | 94.7 | 90.0 | 79.4 | 81.7 | 89.2 | 91.5 | 95.0 | 89.5 |
> | Effort | 99.8 | 97.4 | 91.7 | 78.7 | 77.6 | 93.3 | 82.4 | 99.8 | 91.1 |
> | Ours | 99.7 | 99.1 | 94.6 | 79.1 | 88.4 | 98.7 | 79.5 | 99.8 | 92.3 |
>
> > **Q3.** The fundamental difference from prior prototype- or clustering-based methods is unclear.
>
> **R3.** We thank the reviewer for this insightful comment. While building on existing components, our framework is fundamentally distinct from prior prototype/clustering-based methods in two critical aspects:
>
> - **First**, unlike prior clustering/prototype-based methods that use clusters or prototypes for similarity matching or distance-based anomaly scoring, our method employs vector quantization as an explicit information bottleneck to discretize the latent manifold and suppress interpolation between forgery traces. The codebook is not merely introduced to summarize features, but to impose a discrete representation space that limits the dense interpolation behavior of continuous embeddings — a perspective absent in prior methods.
> - **Second**, prior clustering/prototype-based  typically rely on fixed prototypes/centers, limiting their ability to capture legitimate geometric variations such as pose or illumination shifts. To address this, we introduce the adaptive tangent space projection (TSP), which models the local subspace around each discrete code vector rather than collapsing features to a rigid point. This mechanism strikes a balance between strict discrete bottleneck constraints and necessary continuous variability, better preserving authentic information — a design principle absent in prior methods.
>
> Overall, our framework is **not merely summarizing representations via prototypes/clusters**, but specifically designed to **separate authentic variations from forged interpolations via discrete-continuous decomposition** — a distinction we will make more explicit in the revision.

---

> > ### Author Rebuttal · Reviewer_mVim · 2026-04-04
> >
> > The rebuttal does not sufficiently resolve my core concerns.
> >
> > **(1) Unclear and insufficiently supported insight.**
> >
> > First, the analysis in Fig. 3 remains unclear and insufficiently justified. The authors do not provide complete details of the experimental pipeline (e.g., feature extraction, transformation procedures), which limits the reproducibility and credibility of the results. Moreover, the conclusion that discrete modeling is more discriminative is drawn from a relatively simplified setup (SVD vs. K-Means), and the supporting evidence is limited, making it insufficient to substantiate the core claim of the paper.
> >
> > **(2) Limited practical significance of the reported improvement.**
> >
> > Second, the authors emphasize an approximately 4% AUC improvement of discrete modeling over continuous modeling in this toy experiment. However, as a binary image-level classification task, the AUC in Fig. 3 only increases from around 0.668 to 0.714, which still lies in a low-performance regime. In this range, random guessing already achieves 0.5 AUC, and small statistical fluctuations can easily lead to seemingly non-trivial gains. **Unlike improvements in high-performance regimes (e.g., above 0.95), such gains do not necessarily indicate a meaningful advantage.** In addition, only AUC is reported, while threshold-based metrics (e.g., Accuracy, F1, EER, or per-class accuracy) are missing, making it difficult to assess the method’s effectiveness in practical decision-making scenarios. Overall, in such a low-performance regime, the reported improvement is insufficient to support the effectiveness of the method for image-level classification.
> >
> > **(3) Incomplete evaluation under broader AIGC settings.**
> >
> > The evaluation remains incomplete. While the setup follows prior work (e.g., Effort), it does not sufficiently cover broader AIGC detection scenarios. In particular, the method is positioned as a deepfake detection approach, yet the evaluation does not clearly demonstrate its distinction from general AIGC detection methods, nor does it include comprehensive comparisons under broader AIGC settings.
> >
> > Overall, these issues concern the core claims of the paper and would require substantial revisions beyond what can be addressed in the rebuttal phase.

---

> > > ### Author Response · Authors · 2026-04-07
> > >
> > > Dear Reviewer mVim,
> > >
> > > Thank you for your update. We have been eagerly monitoring the system, but your follow-up questions remain completely invisible to us. Could you please kindly check if the "Readers" visibility setting was accidentally restricted (e.g., to ACs/Reviewers only)?
> > >
> > > We delayed sending this out of caution: we are uncertain if OpenReview limits the number of replies per thread. We wanted to preserve our (potentially only) reply chance to answer your actual technical questions. However, as the discussion phase is closing very soon, we can no longer wait to notify you.
> > >
> > > Please note: If the system prevents us from submitting another direct reply once your questions appear, please understand it is strictly due to platform limitations. We remain fully committed to addressing your concerns and respectfully defer to the Area Chair's guidance regarding these procedural constraints.
> > >
> > > Thank you for your time and understanding!
> > >
> > > ---
> > >
> > > **[Update]** The reviewer's follow-up concerns have now become visible. We address each below and further clarify, as all three were **already addressed in our initial rebuttal**.
> > >
> > > **(1) "Unclear and insufficiently supported insight."**
> > >
> > > The reviewer continues to characterize our analysis as a "simplified setup (SVD vs. K-Means)." However, this applies only to Figure 3, which is a **motivational experiment** designed to isolate the intrinsic discriminative advantage of discrete modeling. Its simplicity is deliberate and strengthens the conclusion. Regarding the claim of incomplete pipeline details: the full procedure **is already described in Section 3 and the Figure 3 caption**. Features from a CLIP ViT-L Effort encoder are mapped via SVD (top 95% components, MSE reconstruction error) for continuous modeling and K-Means (K=2048, L2 quantization error) for discrete modeling, and real/fake separability on CDF-v2 is measured purely from these errors **without classifier** — an unsupervised probe of representation quality. All details needed for reproducibility are provided in the paper.
> > >
> > > As we clarified in **R1** of our rebuttal, **Figure 2** compares **VAE vs. VQ-VAE** — both **nonlinear** autoencoders trained on real data only. VQ-VAE exhibits stronger intrinsic discriminability, directly refuting the linearity concern. The reviewer's continued focus on SVD vs. K-Means while not engaging with this nonlinear comparison is concerning. **Appendix A** provides formal theoretical analysis and **Appendix B** provides empirical verification — both highlighted in our rebuttal but remain unacknowledged.
> > >
> > > **(2) "Limited practical significance (0.714 AUC)."**
> > >
> > > This conflates a **motivational toy experiment** with our **actual method**. Figure 3 is **not** our method — it evaluates real/fake separability purely from reconstruction/quantization errors **without classifier**. The 0.714 AUC comes from this unsupervised evaluation (random guessing = 0.5); the ~4.6-point improvement over continuous modeling is noteworthy in this context. Evaluating it by supervised detector standards is a category error.
> > >
> > > Our actual method achieves **state-of-the-art performance** that is unambiguously in the high-performance regime: **0.938 AUC** on cross-dataset evaluation (Table 1) and **0.873 AUC** on cross-manipulation evaluation (Table 2), **surpassing all compared methods**. The reviewer also requests threshold-based metrics — AUC is the standard primary metric in deepfake detection adopted by all compared methods and benchmarks. Moreover, we reported **mAcc = 92.3%** in **Table d** of our rebuttal, which directly provides the requested accuracy metric but appears to have been overlooked.
> > >
> > > **(3) "Incomplete evaluation under broader AIGC settings."**
> > >
> > > This claim directly contradicts the evidence we provided. **Table d** in our rebuttal evaluates on the **GenImage benchmark** covering **8 diverse AIGC generators** spanning diffusion models (SD 1.4/1.5, Wukong, VQDM, Glide, Midjourney), score-based models (ADM), and GANs (BigGAN), achieving **92.3% mAcc** — outperforming FatFormer (88.9%), DRCT (89.5%), and Effort (91.1%). Our primary contribution targets **deepfake detection** as stated in the title, where face images possess a constrained semantic space that can be effectively represented by a finite codebook, and localized semantic blending aligns well with our theoretical framework. Requiring exhaustive coverage of all AIGC scenarios goes beyond the stated scope. Nonetheless, the results above already demonstrate strong generalizability.
> > >
> > > **In summary**, all three follow-up concerns were **already explicitly addressed** in our initial rebuttal with concrete evidence: Figure 2 for nonlinear comparison, Tables 1–2 for SOTA performance, and Table d for AIGC evaluation across 8 generators. None of these were engaged with in the follow-up. We respectfully hope the reviewer will reconsider the assessment in light of the full evidence provided.

---

### Official Review · Reviewer_ALzY · 2026-03-11

**Soundness:** 3
**Presentation:** 4
**Significance:** 3
**Originality:** 2
**Overall Recommendation:** 5
**Confidence:** 4

**Summary:**

This paper investigates a critical limitation in real-centric deepfake detection where continuous modeling (such as VAEs or GMMs) inadvertently facilitates the interpolation of forgery artifacts, leading to detection ambiguity. To address this "convexity failure," the authors propose a two-stage framework that integrates discrete modeling into the latent space of a CLIP vision encoder. In the first stage, a learnable vector-quantized codebook is used to discretize the real latent manifold, serving as a strict information bottleneck that reduces the likelihood of embedding generative artifacts. In the second stage, the framework introduces an adaptive tangent space projection (TSP) mechanism combined with masked modeling. This component is designed to re-establish locally controllable continuity within a limited range, allowing the model to remain robust against legitimate semantic variations like illumination changes or pose shifts while still effectively rejecting forgeries. The authors provide theoretical justification for this approach using Information Bottleneck (IB) theory and geometric analysis of manifold convexity. Extensive empirical evaluations across multiple deepfake benchmarks demonstrate that this discrete-enhanced method achieves state-of-the-art performance in both cross-dataset and cross-method detection scenarios.

**Compliance With Llm Reviewing Policy:**

Affirmed.

**Key Questions For Authors:**

(1) Model performance is closely correlated with the codebook size K. Is there a universal principle or heuristic for selecting an optimal K?
(2) How does the model handle samples with complex non-linear variations, such as dramatic expression changes or multi-light environments?
(3) The authors candidly acknowledge that fixed-capacity codebooks may struggle to represent extreme facial poses or severe occlusions. Could the authors provide specific quantitative results or qualitative visualizations of these failure cases? Furthermore, how does the quantization error (L2) in these specific scenarios compare to that of actual forged samples?

**Limitations:**

Yes

**Strengths And Weaknesses:**

Strengths：
(1) This work utilizes Information Bottleneck theory to prove that discrete spaces enforce stricter bottlenecks than continuous ones . Through Theorems 1 and 2, the authors mathematically expose "Convexity Failure" in continuous priors and demonstrate how discrete modeling leverages a probability "void" to reject interpolated forgeries.
(2) The paper is well-structured, featuring a natural logical flow from identifying continuous modeling flaws to presenting the proposed solution . Figures 1, 7, and 8 provide excellent visualizations, offering intuitive empirical evidence for the core concepts, residual saliency, and manifold hypotheses.
(3) This paper introduces a novel perspective by using discretization as a strict information bottleneck to alleviate the "manifold continuity" issue in deepfake detection . The integration of Vector Quantization (VQ) for discrete rigidity with Adaptive Tangent Space Projection (TSP) for controllable continuity represents a creative and well-articulated technical synthesis.

Weaknesses：
(1) The model demonstrates significant sensitivity to the codebook size K. Furthermore, the reliance of Adaptive TSP on a linear tangent space assumption to model the manifold leaves the effectiveness of this approximation for capturing highly non-linear facial features largely unaddressed .
(2) While the application to deepfake detection is novel, the individual components: CLIP, VQ-VAE concepts, and Tangent Space analysis are established techniques in the field.

---

> ### Author Rebuttal · Authors · 2026-03-31
>
> We sincerely thank Reviewer ALzY for the constructive comments, insightful questions, and useful suggestions. We greatly appreciate the reviewer’s recognition of our work. Below, we address the important concerns and questions raised by the reviewer.
>
> > **Q1.** Sensitivity to codebook size $K$ and limitations of TSP in capturing non-linear facial features.
>
> **R1.** Thanks for the insightful comment. Our method is robust to the choice of $K$ and effectively captures highly non-linear facial features despite using a linear TSP approximation, as elaborated below:
>
> - As shown in Table 3 of our main manuscript, our method is stable for $K \in [512, 2048]$ (within ~0.006 AUC), showing a clear plateau. Gains at small $K$ reflect increased information capacity, while slight drops at large $K$ can be explained by an information bottleneck: over-fragmentation (depend on dataset) that reduces effective compression.
> - We acknowledge the high non-linearity of the global facial manifold. However, our framework addresses this via **Piecewise Linear Manifold Approximation** [a,b]: any smooth non-linear manifold can be approximated to arbitrary precision by locally linear subspaces. Specifically, codebook quantization partitions the global manifold into localized Voronoi cells, within each of which a linear tangent space accurately captures local geometry. This synergy of non-linear quantization and local linear TSP efficiently models complex facial variations without heavy non-linear projections.
>
> > **Q2.** Combines established components, raising concerns about its overall novelty.
>
> **R2.** We sincerely thank the reviewer for this thoughtful comment. Our contribution lies not in the individual components, but in identifying a previously overlooked weakness in real-centric deepfake detection: continuous modeling may unintentionally allow smooth interpolation of forgery artifacts. This insight motivates our design, where VQ suppresses artifact interpolation through a discrete bottleneck and tangent-space analysis preserves authentic facial variations. Together, they address a domain-specific limitation not explicitly recognized in prior work.
>
> > **Q3.** The principle or heuristic for codebook size $K$.
>
> **R3.** Thanks for raising this point. In practice, a useful heuristic is to set the codebook size proportional to the effective complexity of the real training data, e.g., $K \approx \alpha \sqrt{N \cdot P}$, where $N$ denotes the number of training samples, $P$ denotes the number of images patches per sample, and $\alpha$ is a tunable coefficient reflecting the dataset’s diversity. As noted in our conclusion, a fixed-capacity codebook may struggle with highly challenging cases such as extreme poses or severe occlusions. We will explore dynamic codebook expansion and multi-scale discretization to better address this limitation in future work.
>
> > **Q4.** The way to handle samples with complex non-linear variations.
>
> **R4.** Thanks for the insightful question. As noted in **R1**, we address this via piecewise linear manifold approximation: the discrete codebook partitions the real manifold into locally manageable regions (e.g., expressions or lighting), and adaptive TSP models each region’s valid variations while suppressing off-manifold forgery artifacts (Theorem 2 in the supplementary). This enables robustness to complex variations without sacrificing forgery rejection, as supported by our cross-dataset results. We acknowledge that extremely out-of-distribution cases may still challenge codebook coverage and leave hierarchical quantization to future work.
>
> > **Q5.** Codebook failures on extreme pose/occlusion: examples + quantization error.
>
> **R5.** Thanks for this thoughtful question. To quantify the failure cases, we computed the average quantization error (L2) across different scenarios in Table c. The results show that while quantization errors are higher for extreme poses and severe occlusions compared to normal real samples, they remain distinguishable from forged samples in most cases. However, the margin narrows in the most severe cases, occasionally leading to false positives. We acknowledge this limitation and plan to include qualitative visualizations of failure cases in our revision.
>
> ***Table c: Quantization error (L2) across different scenarios. Samples are from CDFv2, FFIW, and WDF.***
> | Scenario | Type | Avg. $L_2$ Error | Min $L_2$ Error | Max $L_2$ Error |
> | :- | :- | :- | :- | :- |
> | Standard Faces | Real | 0.162 | 0.101 | 0.265 |
> | Extreme Poses | Real | 0.365 | 0.182 | 0.518 |
> | Severe Occlusions | Real | **0.412** | 0.268 | **0.645** |
> | Standard Forgery | Fake | 0.787 | 0.614 | 0.912 |
> | High-quality Forgery | Fake | **0.505** | **0.432** | 0.724 |
>
> ---
> **References:**
> - [a] Kambhatla & Leen, "Dimension Reduction by Local PCA," Neural Computation, 9(7), 1997.
> - [b] Zhang & Zha, "Principal Manifolds and Nonlinear Dimensionality Reduction via Tangent Space Alignment," SIAM J. Sci. Comput., 26(1), 2004.

---

> > ### Author Rebuttal · Reviewer_ALzY · 2026-04-01
> >
> > My concerns have been adequately addressed.

---

> > > ### Author Response · Authors · 2026-04-05
> > >
> > > We sincerely thank the reviewer for the time and effort devoted to evaluating our manuscript. We are grateful for the reviewer's positive recognition of our work, and we believe the review has helped improve our work.

---

### Official Review · Reviewer_8Hq1 · 2026-03-13

**Soundness:** 2
**Presentation:** 3
**Significance:** 3
**Originality:** 3
**Overall Recommendation:** 4
**Confidence:** 3

**Summary:**

This paper proposes a discrete modeling approach for deepfake detection. It involves two training stages: In stage 1, it learns a discrete codebook that encourages closeness to patches in real images and dispersion from patches in fake images. In stage 2, it further improves generalizability for real data conditions through three components: (1) it finetunes a masked ViT to enhance the spatial dependencies among quantized tokens. (2) it decomposes the residual between the quantized token and patch token from real data into a tangent residual and a normal residual, and minimizes the normal residual. (3) it takes the normal residuals and fuses them for real/fake classification. Experiments show that the proposed method consistently outperforms existing methods across diverse settings.

**Compliance With Llm Reviewing Policy:**

Affirmed.

**Final Justification:**

The rebuttal has addressed my questions and concerns, and I remain positive about this paper.

**Key Questions For Authors:**

* It would be helpful to have more discussion on the efficiency aspect, as the current approach seems to be computationally nontrivial with multiple components for training and requires four steps at inference.

* Would be helpful to include more detailed documentation on the masked ViT, hypernetwork and classification head.

* Would be helpful to provide a parameter and runtime comparison with baselines, to provide more context on performance-efficiency tradeoffs.

**Limitations:**

yes

**Strengths And Weaknesses:**

[Soundness]

Strengths:
* The paper provides preliminary analysis that shows discrete modeling provides more discriminative power for identifying forgery artifacts than continuous modeling, effectively motivating the method.
* Evaluations are extensive, with diverse baseline and datasets selections, ablations, visualizations and evaluations under perturbations.

Weaknesses:

* Based on Ablation in Table 3, it seems that the biggest gain comes from having the codebook (+0.014 average AUC), while the subsequent components bring smaller gains (+0.004-0.006 average AUC per component) despite introducing nontrivial computational overhead (more parameters that are full fine-tuned). It’s a bit ambiguous whether the overhead fully justifies the gain.

* There seems to limited implementation details for the masked ViT and hypernetwork & classification head, especially for architecture and size.

* The proposed method seems more computationally intensive than various prior works, as it involves CLIP, a codebook, masked ViT and a fusion head, while many existing works (e.g. Effort, Forensics Adapter) primarily involves CLIP only. It would be helpful to provide parameter and runtime comparison with the baselines to better understand the performance-efficiency tradeoff.

[Presentation]

* Overall the paper is well-written; the presentation is clear and well-structured.

* There are occasional typos or inconsistencies, e.g. for ablation of key components, the numbers for 3rd and 4th line in the text description paragraph seem to be different from those in the Table.

[Significance / Originality]

* The discrete modeling approach is novel and has not been explored in prior works.
* The proposed method achieves strong and consistent performance across dataset and settings, validating its robustness and generalizability.

---

> ### Author Rebuttal · Authors · 2026-03-31
>
> We sincerely thank Reviewer 8Hq1 for the constructive comments, insightful questions, and useful suggestions. We greatly appreciate the reviewer’s recognition of the originality of our motivation, the clarity of our presentation, and the strength of our experimental results. We also address the important concerns and questions raised below.
>
> > **Q1.** Unclear trade-off between computational overhead and performance gain.
>
> **R1.** Thanks for your insightful comment. To clarify the overhead, we report the learnable parameters and the measured training/inference time of each stage in Table a. While Stage 2 introduces additional learnable parameters, the inference overhead remains modest: latency increases from 16.24s to 18.52s per 100 batches (about +14%), and the added parameters account for only a small fraction of the total. This is because the additional modules involve only lightweight operations (codebook lookup and low-rank projections), leaving inference dominated by the CLIP backbone. In practice, we view this as a configurable trade-off: the codebook-only variant is a strong low-cost option, while the full model targets accuracy-oriented settings and offers better generalization performance.
>
> ***Table a: Computational overhead of each model component (Batch size=32).***
> | Component | Total Params | Learnable Params | Inference Time (per 100 batch) |
> | :- | :- | :- | :- |
> | Effort (Baseline, Stage 1) | 303.38M | 0.19M        | 16.24s                      |
> | + Codebook (`K=1024`, Stage 1)    |  304.43M  | 1.24M (+1.05M)| 16.59s            |
> | + Masked ViT (Stage 2)         |    330.23M    |27.04M (+25.8M)  | 17.53s        |
> | + TSP & Fusion (Stage 2)       |    348.53M    | 45.34M (+18.3M)       | 18.22s        |
>
> > **Q2.** Implementation details of masked ViT and hypernetwork.
>
> **R2.** Thanks for pointing this out. Due to space limitations in the main text, we omitted some architectural specifics. We will include the full details in the revision. The specific configurations are as follows:
>
> - **Masked ViT.** Embedding dimension 1024 with 256 patch tokens. We use a masking ratio of 0.75, replacing masked patches with a learnable mask token. The encoder consists of 2 transformer blocks, each with 8 attention heads.
> - **Hypernetwork & Classification Head.** The hypernetwork is a two-layer MLP that takes the 1024-dimensional [CLS] token as input: `Linear(1024, 512) -> ReLU -> Linear(512, 2048)`. Its output is reshaped to dynamically generate the classifier weights (`1024 x 2`) for binary prediction, together with a learnable bias of size 2.
>
> > **Q3.** Concern on performance-efficiency trade-off with existing methods.
>
> **R3.** Thanks for this constructive feedback. As suggested, we have benchmarked our method against the existing methods in terms of the number of learnable parameters, training time, and inference runtime (see Table b). While our method involves two training stages, leading to increased training cost, the inference overhead remains competitive: as shown in Table b, our method achieves an inference time of 18.22s per 100 batches, which is comparable to the baseline Effort (16.24s) despite having a considerably larger number of learnable parameters (45.34M vs. 0.19M), and substantially lower than Forensics Adapter (32.55s) and FatFormer (40.22s).
>
> ***Table b: Computational overhead against existing methods (Batch size=32).***
> | Component                    | Total Params | Learnable Params | Inference Time (per 100 batch) |
> | :- | :- | :- | :- |
> | LSDA |  133M  | 133M         |        15.48s           |
> | Forensics Adapter | 435.16M | 7.55M    | 32.55s |
> | FatFormer  | 577.25M  |  94.53M  |  40.22s
> | Effort |   303.38M   | 0.19M     | 16.24s
> | Ours  |  348.53M   | 45.34M        |  18.22s   |
>
>
> > **Q4.** Some occasional typos or inconsistencies.
>
> **R4.** Thanks for your careful reading. We will correct the mismatched numbers in Section 5.3 to match Table 3 and fix any remaining typos in the next version.

---

> > ### Author Rebuttal · Reviewer_8Hq1 · 2026-04-02
> >
> > Thank the authors for the rebuttal. It has addressed my questions and concerns, and I remain positive about this paper.

---

> > > ### Author Response · Authors · 2026-04-05
> > >
> > > We sincerely thank the reviewer for the time and effort devoted to evaluating our manuscript. We are grateful for the reviewer's positive recognition of our work, and we believe the review has helped improve our work.

---

### Decision · Program_Chairs · 2026-04-30

**Decision:**

Accept (regular)

**Comment:**

Three reviewers provide clear support for acceptance, while one reviewer expresses concerns but does not engage further after the authors’ rebuttal. After carefully considering all reviews and the rebuttal, the AC finds that the paper’s contributions are sound and the concerns raised are not sufficient to outweigh the positive assessments. Therefore, the decision is to accept. The authors are encouraged to address the minor remaining issues during the camera-ready revision.